usepackagexcolor

# ROBUST DECENTRALIZED VFL
# OVER DYNAMIC DEVICE ENVIRONMENT

## ABSTRACT

Robust collaborative learning on a network of edge devices, for vertically split datasets, is challenging because edge devices may fail due to environment conditions or events such as extreme weather. The current Vertical Federated learning (VFL) approaches assume a centralized learning setup or assume the active party or server cannot fail. To address these limitations, we first formalize the problem of VFL under dynamic network conditions such as faults (named DN-VFL). Then, we develop a novel DN-VFL method called **M**ultiple **A**ggregation with **G**ossip Rounds and **S**imulated Faults (MAGS) that synthesizes faults via dropout, replication, and gossiping to improve robustness significantly over baselines. We also theoretically analyze our proposed approaches to explain why they enhance robustness. Extensive empirical results validate that MAGS is robust across a range of fault rates—including extreme fault rates—compared to prior VFL approaches.

## 1 INTRODUCTION

Collaborative cross-device learning and inference on IoT or edge devices present unique challenges not encountered in cross-silo setups (Yuan et al., 2023), such as limited power resources, device unreliability, and the absence of a centralized server. Particularly, when the application requires devices to collaborate in predicting a global feature of the environment, these challenges become critical. For example, deploying a network of sensors for intelligent monitoring in harsh environments (e.g., deep sea sensors, underground mines, or remote rural areas) involves devices that may fail due to power constraints or extreme weather conditions. Additionally, internet connectivity may be limited or non-existent, and no single device can be considered a perfectly reliable central server. Yet, in safety-critical applications such as search and rescue in underground mines, this intelligent device network needs to continue operating even under near catastrophic faults (e.g., 50% of devices fail). Therefore, in this work, we seek to answer the following: **Can we develop a cross-device collaborative method that maintains strong performance at test time even under near-catastrophic faults in the decentralized setting?**

Vertical Federated Learning (VFL) (Liu et al., 2024) emerges as a natural solution for tasks requiring device collaboration at inference time. In VFL, clients share the same set of samples but have different features. In our environmental monitoring example, the samples correspond to unique timestamps, and the features correspond to sensor data from each device—each providing a partial view of the global environment. Previous research in VFL has explored aspects of fault tolerance and decentralized learning, primarily focusing on the training phase. For instance, studies have addressed asynchronous communication to handle device failures during training (Chen et al., 2020; Zhang et al., 2021; Li et al., 2020; 2023). An exception is the work by Sun et al. (2023), who proposed Party-wise Dropout to mitigate inference-time faults caused by passive parties (clients) dropping off unexpectedly, but they assumed that the active party (server) remains fault-free. Other works have focused on communication efficiency in decentralized VFL (Valdeira et al., 2023). However, to the best of our knowledge, no prior work simultaneously addresses decentralized learning and arbitrary faults—including the active party or server—during inference. This gap, as summarized in Table 1, motivates our research.

---

[1]Even though Sun et al. (2023) did not explicitly consider client faults during training, the method from Sun et al. (2023) could handle training faults by treating them like Party-wise Dropout as discussed in Section 3.1.

Table 1: Our MAGS method considers the cross-device decentralized VFL setting where faults can occur in both clients and the active party or server, unlike the existing literature in decentralized or fault-tolerant VFL.

| | Context | Decentralized | Faults During | | Fault Types | |
|---|---|---|---|---|---|---|
| | | | Training | Inference | Client | Active Party/Server |
| STCD(Valdeira et al., 2023) | Cross-silo | ✓ | ✗ | ✗ | ✗ | ✗ |
| VAFL(Chen et al., 2020) | Cross-silo | ✗ | ✓ | ✗ | ✓ | ✗ |
| Straggler VFL(Li et al., 2023) | Cross-silo | ✗ | ✓ | ✗ | ✓ | ✗ |
| Party-wise dropout (Sun et al., 2023) | Cross-silo | ✗ | ✓[1] | ✓ | ✓ | ✗ |
| MAGS (ours) | Cross-device | ✓ | ✓ | ✓ | ✓ | ✓ |

To address these challenges holistically, we first formalize this problem setup and then propose a solution, **M**ultiple **A**ggregation with **G**ossip Rounds and **S**imulated Faults (MAGS). We also define our context, including, data assumption, network model, and a measure of performance in this context called Dynamic Risk. A comparison of context presented in this work with vanilla VFL is captured in Figure 1. MAGS significantly improves robustness by integrating three interconnected methods that build upon and complement each other. First, during training, we simulate high fault rates via dropout so that the model can be robust to more missing values at test time. Second, we replicate the data aggregator to prevent catastrophic failure in case the active party (or server) goes down during test time. Third, we introduce gossip rounds to implicitly ensemble the predictions from multiple data aggregators, reducing the prediction variance across devices. Finally, we evaluate the effectiveness of MAGS by conducting experiments using five datasets (StarCraftMNIST (Kulinski et al., 2023) in the main paper and MNIST, CIFAR10, CIFAR100, Tiny ImageNet in the appendix) and different network configurations. The results establish that MAGS is significantly more robust than prior methods, often improving performance more than 20% over prior methods at high fault rates. We summarize our contributions as follows:

- We formalize the problem of decentralized VFL under dynamic network conditions, called Dynamic Network VFL (DN-VFL), and define Dynamic Risk, which measures performance under (extreme) dynamic network conditions.

- We develop and analyze MAGS, that combines fault simulation, replication, and gossiping to enable strong fault tolerance for DN-VFL.

- We demonstrate that MAGS is significantly more robust to dynamic network faults than prior methods across multiple datasets, often improving performance more than 20% compared to prior methods.

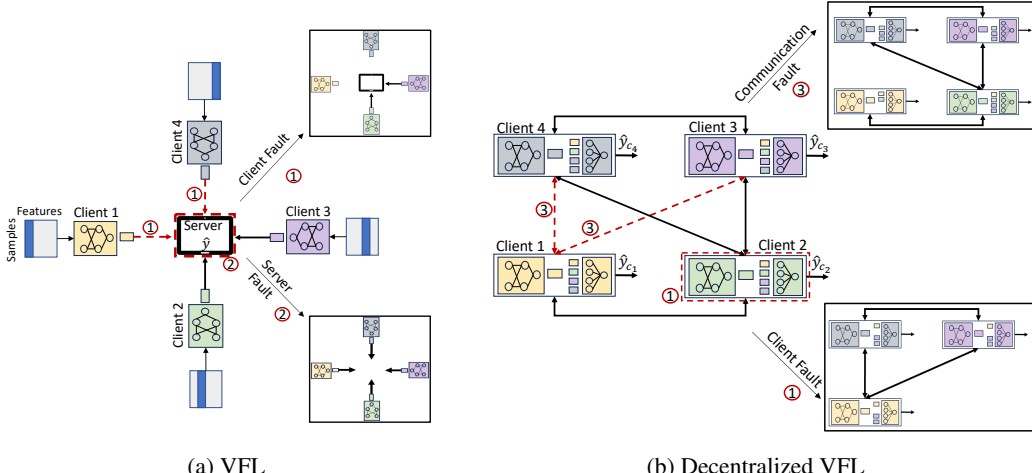

(a) VFL  (b) Decentralized VFL

Figure 1: Vanilla VFL (Figure 1a) assumes samples are split across clients with a central server. The data context in our study is the same as VFL where the features are split across clients. However, in our case, no centralized server node is assumed, and clients serve as data aggregators (Figure 1b). Our goal is to obtain robust test time performance even under highly dynamic networks such as client/device faults (①), server faults (②) and communication faults (③).

## 1.1 RELATED WORKS

**Network Dynamic Resilient FL** In VFL, network dynamics has mostly been studied during the training phase for asynchronous client participation Chen et al. (2020); Zhang et al. (2021); Li et al. (2020; 2023). Research on VFL network dynamics during inference is limited. Ceballos et al. (2020) noted performance drops due to random client failures during testing, and Sun et al. (2023) studied passive parties dropping off randomly during inference and proposed Party-wise Dropout (PD). Thus, prior dynamic network resilient VFL methods have majorly focused on train-time faults and assumed a special node (server or active party) that is immune to failure. As VFL differs significantly from the horizontal FL (HFL) setting (Yang et al., 2019), where clients share the same set of features but have different samples, HFL methods for handling faults (e.g., adaptive aggregation of different models Ruan et al. (2021)) are inapplicable in our scenario. Additionally, unlike HFL, where faults only affect training, faults in VFL can disrupt both training *and* inference due to the need for client communication during inference.

**Decentralized FL** Conventional FL uses a central server for aggregation. However, this approach results in server being the single point of failure. To address such limitations, Decentralized FL has been considered (Yuan et al., 2023). Unlike, the extensively studied HFL decentralized methods (Tang et al. (2022); Lalitha et al. (2019); Feng et al. (2021); Gabrielli et al. (2023)), VFL decentralized methods are limited. For the special case of simple linear models, He et al. (2018) proposed decentralized algorithm COLA. For more general split-NN models, Valdeira et al. (2023) proposed decentralized, STCD, and a semi-decentralized, MTCD, methods. Neither COLA nor STCD/MTCD analyze network dynamics such as faults during inference time.

## 2 PROBLEM FORMULATION

We define a novel formulation, *DN-VFL*, which specifies both an operating context and the desired properties of a learning system. The context is comprised of two entities: the data context and the network context, which we define in the first subsection. The desired property of the system is robustness under dynamic conditions, which we formally define via *Dynamic Risk* and corresponding metrics in the next two subsections.

**Notation** Let $(X \in \mathcal{X}, Y \in \mathcal{Y})$ denote the random variables corresponding to the input features and target, respectively, whose joint distribution is $p(X, Y)$. With a slight abuse of notation, we will use $\mathcal{Y}$ to denote one-hot encoded class labels and probability vectors (for predictions). Let $\mathcal{D} = \{(\boldsymbol{x}_i, y_i)\}_{i=1}^n$ denote a training dataset of of $n$ samples i.i.d. samples from $p(X, Y)$ with $d$ input features $\boldsymbol{x}_i$ and corresponding target $y_i$. Let $\boldsymbol{x}_{\mathcal{S}}$ denote the subvector associated with the indices in $\mathcal{S} \subseteq \{1, 2, \cdots, d\}$, e.g., if $\mathcal{S} = \{1, 5, 8\}$, then $\boldsymbol{x}_{\mathcal{S}} = [x_1, x_5, x_8]^T$. For $C$ clients, the dataset at each client $c \in \{1, 2, \cdots, C\}$ will be denoted by $\mathcal{D}_c$. Let $\mathcal{G} = (\mathcal{C}, \mathcal{E})$ denote a network (or graph) of clients, where $\mathcal{C} \subseteq \{1, 2, \cdots, C\} \cup \{0\}$ denotes the clients plus an entity (possibly an external entity or one of the clients itself) that represents the device that collects the final prediction and has the labels during training (further details in Section 2.2) and $\mathcal{E}$ denotes the communication edges.

## 2.1 DYNAMIC NETWORK VFL CONTEXT

**Data and Network Context** A partial features data context means that each client has access to a subset of the features, i.e., $\mathcal{D}_c = \{\boldsymbol{x}_{i,\mathcal{S}_c}\}_{i=1}^n$, where $\mathcal{S}_c \subset \{1, 2, \cdots, d\}$ for each client $c$. This is the same *data context* as SplitVFL (Liu et al., 2022), a variant of VFL, which incorporates the idea of split learning (Vepakomma et al., 2018), and jointly trains models at both server and clients. Furthermore, in this study we assume that the features with each client is a partition of the feature set of a sample and each client has disjoint set of features for each sample. However, we do allow for scenarios where the clients can have features among one another that are correlated. For instance, there can be two sensors that can have correlated features due to their physical proximity. Unlike vanilla VFL, in DN-VFL we allow the clients to act as data aggregators and communicate with one another. This leads to the realization of Decentralized VFL. Through the rest of the paper, the terms clients and devices are used interchangeably.

**Definition 1** (Dynamic Network Context). *A dynamic network means that the communication graph can change across time indexed by $t$, i.e., $\mathcal{G}(t) = (\mathcal{C}(t), \mathcal{E}(t))$, where the changes over time can be either deterministic or stochastic functions of $t$.*

This dynamic network context includes many possible scenarios including various network topologies, clients joining or leaving the network, and communication being limited or intermittent due to power constraints or physical connection interference. We provide two concrete dynamic models where there are device failures or communication failures. For simplicity, we will assume there is a base network topology $\mathcal{G}_{\text{base}} = (\mathcal{C}_{\text{base}}, \mathcal{E}_{\text{base}})$ (e.g., complete graph, grid graph or preferential-attachment graph), and we will assume a discrete-time version of a dynamic network where $t \in \{0, 1, 2, \cdots\}$, which designates a synchronous communication round. Given this, we can formally define two simple dynamic network models that encode random device and communication faults.

**Definition 2** (Device Fault Dynamic Network). *Given a fault rate $r$ and a baseline topology $\mathcal{G}_{\text{base}}$, a device fault dynamic network $\mathcal{G}_r(t)$ means that a client is in the network at time $t$ with probability $1 - r$, i.e., $\Pr(c \in \mathcal{C}_r(t)) = 1 - r, \forall c \in \mathcal{C}_{\text{base}}$ and $\mathcal{E}_r(t) = \{(c, c') \in \mathcal{E}_{\text{base}} : c, c' \in \mathcal{C}_r(t)\}$.*

**Definition 3** (Communication Fault Dynamic Network). *Given a fault rate $r$ and a baseline topology $\mathcal{G}_{\text{base}}$, a communication fault dynamic network $\mathcal{G}_r^{\text{CF}}(t)$ means that a communication edge (excluding self-communication) is in the network at time $t$ with probability $1 - r$, i.e., $\mathcal{C}_r^{\text{CF}}(t) = \mathcal{C}_{\text{base}}$ and $\Pr((c, c') \in \mathcal{E}_r^{\text{CF}}(t)) = 1 - r, \forall (c, c') \in \mathcal{E}_{\text{base}}$ where $c \neq c'$.*

As this work focuses on the foundations of Dynamic Network VFL, we only experimented with these two dynamic network models. However, more complex dynamic models could be explored in the future. For example, the networks could change smoothly over time (e.g., one connection being removed or added at every time point). Or, a network could model a catastrophic event at a particular time $t'$ followed by a slow recovery of the network as devices are reconnected or restarted. We leave the investigation of more complex dynamic models to future work.

### 2.2 DN-VFL Problem Formulation via Dynamic Risk

Given these context definitions, we now define the goal of DN-VFL in terms of the Dynamic Risk which we define next. For now, we will assume the existence of a distributed inference algorithm $\Psi(\boldsymbol{x}; \theta, \mathcal{G}(t)) : \mathcal{X} \to \mathcal{Y}^C$, where each client makes a prediction across the data-split network under the dynamic conditions given by $\mathcal{G}(t)$. In section 3, we will propose a natural message passing distributed inference algorithm that generalizes VFL. Furthermore, we will use a (possibly stochastic) post-processing function $h : \mathcal{Y}^C \to \mathcal{Y}$ to model the final communication round between the clients and another entity (which may be external or may be one of the clients), which owns the labels for training and collects the final prediction during the test time. The $h$ can model different scenarios including where the entity has access to all or only a single client's predictions. As an example, the entity could represent a drone passing over a remote sensing network to gather predictions or a physical connection to the devices at test time (e.g., when the sensors are ultra-low power and cannot directly connect to the internet). Or, this entity could represent a power intensive connection via satellite to some base station that would only activate when requested to save power.

**Definition 4** (Dynamic Risk). *Assuming the partial features data context (subsection 2.1) and given a dynamic network $\mathcal{G}(t)$ (Definition 1), the Dynamic Risk is defined as: $R_h(\theta; \mathcal{G}(t)) \triangleq \mathbb{E}_{X,Y,\mathcal{G}(t),h}[\ell_h(\Psi(X; \theta, \mathcal{G}(t)), Y)]$, where $\Psi : \mathcal{X} \to \mathcal{Y}^C$ is a distributed inference algorithm parameterized by $\theta$ that outputs one prediction for each client and $\ell_h(\boldsymbol{y}, y) \triangleq \ell(h(\boldsymbol{y}; \mathcal{G}(T)), y)$ is a loss function where $h : \mathcal{Y}^C \to \mathcal{Y}$ (which could be stochastic) post-processes the client-specific outputs to create a single output based on the communication graph at the final inference time $T$, and $\ell$ could be any standard loss function.*

This risk modifies the usual risk by also taking an expectation w.r.t. the dynamic graph (which could be stochastic over time) and a the client selection function $h$, which will be described more below. We assume that the distributed inference algorithm produces a a prediction for every client and the $h$ function (stochastically) selects the final output (note how the composition is a normal prediction function, i.e., $h \circ \Psi : \mathcal{X} \to \mathcal{Y}$). We will use the term "system" or "network" instead of "model" as all computation must be computed in a distributed manner. This means that the network's parameters $\theta$ are distributed across all clients. We also note that the model at each client could have different parameters and even different architectures, unlike in HFL.

Now we will define the final processing function $h$ which represents the communication to the external entity. We consider two practical scenarios and two oracle methods that depend on how $h$ selects the final output of the distributed inference algorithm. These four methods for defining $h$ will

represent Dynamic Risks under different scenarios and form the basis for the test metrics used in the experiments. The output of the inference algorithm forms the basis on which the test accuracy is computed. We first formally define an active set $\mathcal{A}$ of clients at the last communication round as $\mathcal{A}(\mathcal{G}(T)) \triangleq \{c : (0,c) \in \mathcal{E}(T), c \in \mathcal{C}(T)\}$, which means the devices that could communicate to the special entity denoted by $0$ at the last communication round (other devices are not able to communicate their predictions).

**Select Active Client** One natural measure is to use the output of a randomly selected *active* client and if there are no active clients then output the dummy marginal probability of $Y$ (corresponding to a catastrophic failure of all devices), i.e., $\Pr(h_{\text{active}}(\hat{\boldsymbol{y}}) = \hat{y}_c \,|\, |\mathcal{A}| > 0) = \frac{1}{|\mathcal{A}|}, \forall c \in \mathcal{A}$ and $\Pr(h_{\text{active}}(\hat{\boldsymbol{y}}) = p(Y) \,|\, |\mathcal{A}| = 0) = 1$.

**Select Oracle Best and Worst Active Client** We now provide two bounds on selecting the best and worst client in the active set ($\mathcal{A}$). These are oracle functions because they require access to the true label $y$. Intuitively, for oracle best, if any active client prediction is correct, we predict the correct label. Similarly, for oracle worst, if any active client prediction is incorrect, we predict the wrong label. The worst case lower bounds a single client prediction, i.e., the system's accuracy even if the worst client is selected every time. We can formally define these as:

$$h_{\text{best}}(\hat{\boldsymbol{y}}) \triangleq \begin{cases} y, & \text{if } y \in \{\arg\max_j \hat{y}_{c,j} : c \in \mathcal{A}\} \\ y', & \text{otherwise, where } y' \neq y \end{cases} \quad \text{and} \quad h_{\text{worst}}(\hat{\boldsymbol{y}}) \triangleq \begin{cases} y', & \text{if } \exists y' \in \mathcal{A}, y' \neq y \\ y, & \text{otherwise} \end{cases}.$$

**Select Any Client** Finally, a different case is the prediction if a device is chosen at random from all devices both active and inactive. This models the case where the external entity queries a specific device but does not know whether the device can communicate its output or not. If the randomly selected device is not in the active set, then this $h$ will give the dummy prediction of $p(Y)$. Formally, the select any client $h_{\text{any}}$ can be defined as $\Pr(h_{\text{any}}(\hat{\boldsymbol{y}}) = \hat{y}_c) = \frac{1}{C}, \forall c \in \mathcal{A}$ and $\Pr(h_{\text{any}}(\hat{\boldsymbol{y}}) = p(Y)) = \frac{C - |\mathcal{A}|}{C}$.

## 3 MULTIPLE AGGREGATION WITH GOSSIP ROUNDS AND SIMULATED FAULTS (MAGS)

Given the novel DN-VFL context, we now propose our message passing distributed inference algorithm MAGS for DN-VFL and present the relevant theoretical insights. **First**, we extend and discuss dropout methods for simulating faults during training to enhance the robustness of the network with faults at test time. **Second**, we overcome the problem where VFL catastrophic fails if the single aggregator node faults by enabling multiple clients to be data aggregators. **Third**, we improve both the ML performance and decrease the variability of client-specific predictions by using gossip rounds to average the final output across devices. We assume that the aggregator of neighbor representations is simply concatenation and the network architecture are based on simple multi-layer perceptrons (MLP), which is similar to vanilla VFL architectures. We summarize the different proposed techniques and contrasts them with VFL in Figure 3 (in Appendix) and present the MAGS inference algorithm in Algorithm 1.

---

**Algorithm 1** MAGS Inference Algorithm

---

1: **Input:** Input features $\{\boldsymbol{x}_c\}_{c=1}^C$, parameters $\{\theta_c^{(t)} : \forall c, t\}$, and dynamic graph $\mathcal{G}(t) = (\mathcal{C}(t), \mathcal{E}(t))$
2: $\boldsymbol{z}_c^{(0)} = f_c^{(0)}(\boldsymbol{x}_c; \theta_c^{(0)}), \quad \forall c \in \mathcal{C}(0)$                             {Process input at all clients}
3: $\tilde{\boldsymbol{z}}_k^{(1)} = g(\{\boldsymbol{z}_{c'}^{(0)} : (k,c') \in \mathcal{E}(1)\}), \quad \forall k \in \mathcal{K} \cap \mathcal{C}(1)$     {Aggregate messages from neighbors}
4: $\boldsymbol{z}_k^{(1)} = f_k^{(1)}(\tilde{\boldsymbol{z}}_k^{(1)}; \theta_k^{(1)}), \quad \forall k \in \mathcal{K} \cap \mathcal{C}(1)$     {Apply prediction function to aggregated output}
5: **for** $t \leftarrow 2, \ldots, G+1$ **do**                         {Gossip prediction probabilities to neighbors}
6:     $\boldsymbol{z}_k^{(t)} = \text{Avg}(\{\boldsymbol{z}_{k'}^{(t-1)} : (k,k') \in \mathcal{E}(t), k' \in \mathcal{K} \cap \mathcal{C}(t)\}), \quad \forall k \in \mathcal{K} \cap \mathcal{C}(t)$
7: **end for**
8: **return** $\{\boldsymbol{z}_k^{(G+1)} \in \mathcal{Y}\}_{k \in \mathcal{K}}$                     {Return all aggregator-specific predictions}

---

### 3.1 Decentralized Training of MAGS with Real and Simulated Faults via Dropout

To train MAGS, we use a standard VFL backpropagation algorithm *without gossip rounds*[2]., which only requires two communication rounds: one for the forward pass and one for the backward pass. In both passes, faults can be treated similarly to dropout, where missing values are imputed with zeros (see appendix for more details). Our training algorithm assumes all devices have access to the labels, which is similar to an assumption made in Castiglia et al. (2022) and is valid for our setup, involving a trusted but unreliable device network, where robustness is our primary goal. While we aim for robustness against severe, potentially catastrophic faults, we expect a relatively stable and reliable device network during normal training, with a small fault rate (e.g., 1%-5%). However, training solely with a low fault rate may leave the model vulnerable to higher fault rates during inference, which can occur due to external factors like extreme weather. This presents a challenge, as large-scale inference-time faults lead to missing values, causing a distribution shift between the training and test data. Such shifts can severely degrade model performance, as noted by Koh et al. (2021).

A natural way to address this issue is to simulate inference-time faults during training using dropout. Sun et al. (2023) introduced Party-wise Dropout (PD) for server-based VFL, simulating random client failures during communication with the server. However, since DN-VFL operates in a decentralized environment where clients communicate with each other, PD is insufficient. To model this decentralized communication, we propose Communication-wise Dropout (CD), which applies dropout to client-to-client communication instead of just client-to-server communication. PD simulates device failures, while CD simulates communication failures. For further clarity, we provide detailed comparisons between PD and CD configurations in the decentralized setting in the appendix.

To enhance model robustness, we introduce additional dropout beyond what occurs naturally from real network faults, simulating higher fault rates during training. This approach is based on the theoretical understanding of dropout's regularizing effect, as discussed in the literature. Baldi & Sadowski (2013) demonstrated that dropout can act as a regularizer during training. Mianjy & Arora (2020) further showed that a model trained with dropout and tested without it can achieve near-optimal test performance in $O(1/\epsilon)$ iterations. Their work also provides evidence that, in over-parameterized models, dropout-regularized networks can generalize well even when dropout is applied during testing—exactly what is needed in DN-VFL, where faults may occur during both training and inference.

### 3.2 Multiple Aggregators in Decentralized VFL (MVFL)

Because we are in the decentralized setting, a key problem in the conventional VFL setup is that there is a single point of failure, i.e., the single server or data aggregator. Thus, the server going down results in a catastrophic failure and a higher lower bound on $R_h(\theta; \mathcal{G}_r(t))$. Hence, to significantly reduce this system-level failure, we propose the use of all clients as data aggregators to introduce fault-tolerance via redundancy. We call this Multiple VFL (MVFL) for the decentralized VFL setup. In Algorithm 1 lines 3 and 4 denote using multiple data aggregators. An MVFL setup can tolerate the failure of any node and the probability of failure of all nodes is given by $r^C$, which is very small if $C$ is large. However, having all nodes act as aggregators could increase the communication cost. Thus, we develop $K$-MVFL as a low communication cost alternative to MVFL. In $K$-MVFL, we assume there is a set of clients $\mathcal{K} \subseteq \mathcal{C}$ that act as data aggregators. The number of aggregators ($K \triangleq |\mathcal{K}|$) will generally be less than the total number of devices, resulting in lower communication cost than MVFL. We now theoretically prove a bound on the risk that critically depends on the probability of catastrophic failure, i.e., when there are no active aggregators $|\mathcal{A}| = 0$.

**Proposition 1.** *Given a device fault rate $r$, the number of data aggregators $K \leq C$ and the post-processing function $h_{\text{active}}$, and assuming the risk of a predictor (data aggregator) with faults is higher than that without faults, then the dynamic risk with faults is lower bounded by:*

$$\underbrace{R_h(\theta; \mathcal{G}_r(t))}_{\text{Risk with faults}} \geq (1 - r^K) \cdot \underbrace{R_h(\theta; \mathcal{G}_{\text{base}})}_{\text{Risk without faults}} + \underbrace{r^K}_{\Pr(|\mathcal{A}|=0)} \cdot \underbrace{\mathbb{E}[\ell(Y, p(Y))]}_{\text{Risk of random predictor}} . \tag{1}$$

---

[2]By training without gossip, the classifier head on each device is trained independently to maximize its own performance so that its errors are uncorrelated with other devices when using gossiping as an ensembling approach as discussed in Section 3.3

Proof is in the appendix. As a simple application, suppose that the fault rate is very high at $r = 0.3$, this would mean that with VFL 30% of the time the system would fail and the dynamic risk would reduce to random guessing. However, with just four aggregators, the chance of failure reduces to less than 1%. While having multiple aggregators addresses the fundamental problem of catastrophic failures, each model is often insufficient given only one communication round especially for sparse base graphs or high fault rates. Additionally, each device may have widely varying performance characteristics due to its local neighborhood. Thus, further enhancements are needed for robustness and stability.

### 3.3 GOSSIP LAYERS TO ENSEMBLE AGGREGATOR PREDICTIONS

While multiple data aggregators help avoid system-level failures, the performance of each data aggregator may be poor due to faults, which could result in overall high dynamic risk even if catastrophic failures are alleviated. Because gossip is not used during training, each of data aggregator model is different because each will have access to different client representations due to the graph topology and faults at test time (see "Active Worst" metric). Additionally, this variability between data aggregators translates to inconsistent performance when viewed by an external entity as it depends on which device is selected and the best device may differ for each inference query (see the difference between "Active Worst" and "Active Best" metrics). Thus, we propose to use gossip layers to combine predictions among data aggregators and in Algorithm 1, lines 5 and 6 denotes how it is accomplished algorithmically. From one perspective, gossip implicitly produces an ensemble prediction at each aggregator, which we prove always has better or equal dynamic risk. From another perspective, gossiping will cause the aggregator predictions to converge to the same prediction—which means that the system performance will be the same regardless of which device is selected.

To analyze the ML performance of gossip, we leverage the formalization of ensemble diversity related to risk as developed in Wood et al. (2023). Wood et al. (2023) showed that ensemble diversity can be conceptually viewed as another dimension to the bias-variance risk decomposition. In particular, Wood et al. (2023) showed that the ensemble risk can be decomposed into individual risks minus a diversity term (which will reduce the risk if positive). We leverage this theory to prove that the dynamic risk of our ensemble is equal to the non-ensemble dynamic risk minus a diversity term—which is always non-negative and positive if there is any diversity in prediction. This proposition shows that gossiping at inference time, which implicitly creates ensembles, will almost always improve the dynamic risk compared to not using gossip. (Proof is in appendix.)

**Proposition 2.** *The dynamic risk of an ensemble over aggregators is equal to the non-ensemble risk minus a non-negative diversity term:*

$$R_h^{\mathrm{ens}}(\theta; \mathcal{G}(t)) = \underbrace{R_h(\theta; \mathcal{G}(t))}_{\text{Non-ensemble risk}} - \underbrace{\mathbb{E}_{\boldsymbol{x}, \mathcal{G}(t), h}[\tfrac{1}{K} \textstyle\sum_{k=1}^{K} \ell_h(\Psi_k(\boldsymbol{x}; \theta), \Psi^{\mathrm{ens}}(\boldsymbol{x}; \theta))]}_{\text{Diversity term (non-negative)}} \leq R_h(\theta; \mathcal{G}(t)),$$

*where $R_h^{\mathrm{ens}}$ is the ensemble risk; $\Psi_k$ is the $k$-th aggregator model; $\Psi_{\mathrm{ens}}$ is the ensemble model where $\Psi_k^{\mathrm{ens}}(\boldsymbol{x}; \theta, \mathcal{G}(t)) \triangleq Z^{-1} \exp(\sum_{k'=1}^{K} \Psi_{k'}(\boldsymbol{x}; \theta, \mathcal{G}(t))), \forall k \in \mathcal{K}$, where $Z$ is the normalizing constant; and where the notational dependence of $\Psi$ on $\mathcal{G}(t)$ is suppressed and $\ell_h$ in the diversity term applies $h$ to both loss arguments with a slight abuse of notation.*

To analyze output variability using gossip, we turn to gossip consensus results. We first introduce some additional notation. Let $A$ be the adjacency matrix of graph $\mathcal{G} = \{\mathcal{C}, \mathcal{E}\}$ and let $V = D^{-1}A$ denote the consensus matrix where $D$ is the degree matrix (note that $V$ is row stochastic) and let $\lambda$ denote the largest eigenvalue of $V - \frac{\mathbf{1}\mathbf{1}^T}{C}$, also known as the spectral radius. The result below proves that with simple averagning the variability decreases exponentially with increasing gossip rounds based on the spectral radius of the (faulted) graph.

**Proposition 3.** *If simple averaging is used during gossip, the difference between the average output over all devices, denoted $\bar{y}$, and the original output of the $i$-th device, denoted $y_i$, after $G$ gossip rounds is bounded as follows: $\|\bar{y} - y_i\|_2 \leq \lambda^G \sqrt{C} \max_{j,j' \in \mathcal{C}} \|y_j - y_{j'}\|_2, \forall c \in \mathcal{C}$.*

Proposition 3 follows as a special case of the proof in Lin et al. (2021). Thus, assuming the graph is connected (i.e., $\lambda < 1$), the variability between aggregators shrinks to zero exponentially w.r.t. the number of gossip rounds $G$ based on the spectral radius $\lambda$. Intuitively, the spectral radius is small for dense graphs and large for sparse graphs. As a consequence, test-time faults will make the spectral

radius increase. However, as long as the graph is still connected, this gossip protocol can significantly reduce device variability even after only a few gossip rounds.

## 4 EXPERIMENTS

**Datasets**    To cover a diversity of datasets, we test with MNIST, StarCraftMNIST (Kulinski et al., 2023), CIFAR10, CIFAR100 and Tiny ImageNet. StarCraftMNIST (Kulinski et al., 2023) is a spatial reasoning dataset constructed from replays of humans playing the StarCraft II real-time strategy game. Due to space constraints, we only show results for StarCraftMNIST as it is specifically designed to study tasks over a geospatial sensor network, which matches with the context described in the use-cases (Section 1). To simulate a sensor network grid, we split the images into a grid of patches and assign one client to each patch. We mainly present results using 16 clients in a 4x4 grid for StarCraftMNIST. Additional results with different datasets and different numbers of devices can be found in Appendix G.

**Method**    Baseline methods are vanilla *VFL* and VFL with partywise dropout (*PD-VFL*) from Sun et al. (2023). We then include various versions of our MAGS to show the importance of each component to robust DN-VFL performance. Specifically, *MVFL* refers to using all clients as aggregators. *4-MVFL* refers to the low communication version of MVFL, where 4 was chosen based on Proposition 1. The prefix of *PD-* or *CD-* refers to using party-wise or communication-wise dropout during training. And the suffix of *-Gg* denotes that $g$ gossip rounds were used. See Appendix for specifics about model architecture and hyperparameters.

**Baseline Communication Network**    We consider a diversity of graph types and levels of sparsity including the dense complete graph, a grid graph, and a sparse ring graph. We also consider a random geometric graph that generalizes the grid graph such that all devices within a specified distance are connected. We assume a synchronous communication model as is standard in most FL and VFL works (e.g., McMahan et al. (2017); Wang et al. (2022b); Crawshaw et al. (2024); Li et al. (2023); Jiang et al. (2022)).

**Fault Models**    We compute dynamic risk under both device faults and communication faults defined in Definition 2 and Definition 3, respectively. We investigate a wide range of fault rates up to 50% faults, which showcases the method's performance under extreme fault scenarios. Here we present results such that the faulted graph remains constant through duration of an inference. In appendix, results are presented with temporally varying inference fault model.

**Different Test Fault Rates and Patterns**    As seen in Figure 2, across multiple test fault rates, fault types, and baseline networks, the performance of most approaches degrades significantly from about 80% to 30% while our proposed methods (PD-MVFL-G4 and CD-MVFL-G4) are relatively resilient to the increasing fault percentage. By comparing MVFL to VFL, it appears that using multiple data aggregators improves resilience to faults. This observation is in line with Proposition 1 and substantiates the benefit of having multiple data aggregators to deal with DN-VFL. Furthermore, dropout during training leads to improved resilience. MVFL models trained using PD or CD are more robust than MVFL. Such result provides empirical evidence to support the claim that simulating training fault via dropout is a valuable technique to handle inference faults.

The gossip variants of the proposed methods provide a performance boost when combined with PD and CD variants across different fault rates. This underscores the importance of using gossiping as a part of MAGS. The performance of CD/PD-4-MVFL-G2 in Figure 2, indicates that better robustness to inference faults than VFL can be achieved at a much lower communication cost than that of MVFL. In summary, the combined effect of having a decentralized setup, gossiping and dropout clearly outperforms other methods. In the Appendix we present some more investigation (Ablation Study) on number of gossip rounds and different dropout rates for CD and PD. In addition, an extended version of Figure 2 can be found in the Appendix.

**Communication and Performance Analysis**    To study the trade-off between communication and accuracy, Table 2 presents the performance and approximated number of communication (# Comm.) for different baseline networks. An extended version of Table 2 can be found in the Appendix. Across varying level of graph sparseness, using a decentralized setup with gossiping improves performance by at least 10 percent points and can be as high as 32 percent points when compared to VFL. As expected, this significant improvement in performance comes at the cost of

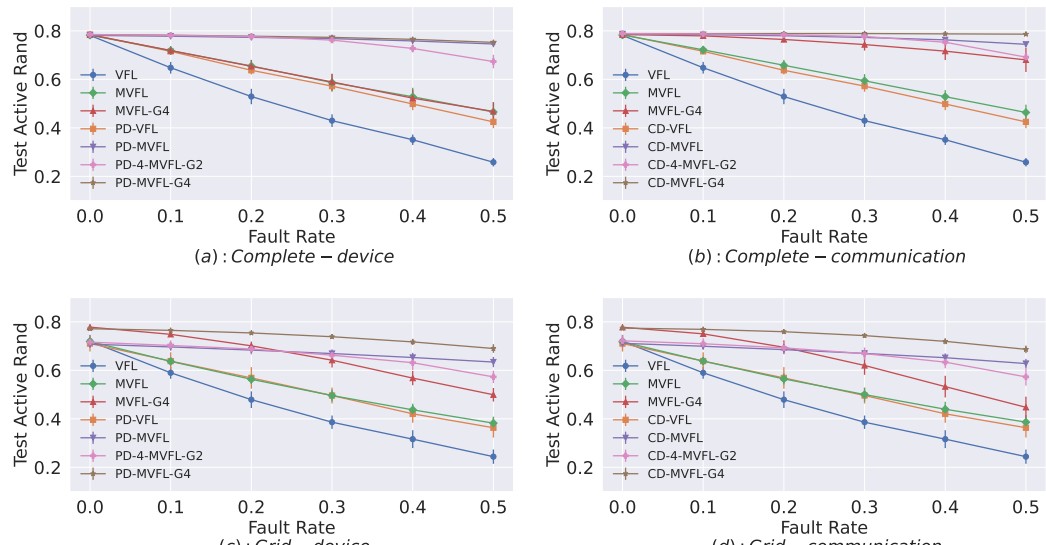

Figure 2: Test accuracy with and without communication (CD-) and party-wise (PD-) Dropout method for StarCraftMNIST with 16 devices. Here we include models trained under an dropout rate of 30% (marked by 'PD-' or 'CD-'). All results are averaged over 16 runs, and the error bar represents standard deviation. Across different configurations, MVFL-G4 trained with feature omissions has the highest average performance, while vanilla VFL performance is not robust as fault rate increases. As our experiments are repeated multiple times, what we report is the expectation (Avg) over the random active client selection.

higher communication, which enables redundancy in the system. Nonetheless, from the results of 4-MVFL, it can be concluded that major improvement over VFL is achieved by just having 4 devices acting as aggregators. Comparing the performance and communication cost of 4-MVFL and MVFL, reveals an efficient trade-off between robustness and communication cost. The ideal setting for the number of aggregators will depend on the context and the cost of a loss in performance. Furthermore, 4-MVFL with a poorly connected graph is still better than VFL with well connected graph, such as 4-MVFL with *RGG* (r=1) versus VFL with *Complete*, where the number of communications are similar in magnitude. This indicates that given a fixed communication budget, VFL may not be the best solution despite its low communication cost. Additionally, we observed reduced impact from gossip communication in sparser networks, implying that increased communication does not always lead to improved performance in dynamic environments.

Table 2: Active Rand (Avg) performance at test time with 30% communication fault rate. Compared to VFL, MVFL performs better but it comes at higher communication cost. Thus we propose 4-MVFL, which is shown to be a low communication cost alternative to MVFL.

|  | Complete | | RGG r=2.5 | | RGG r=2 | | RGG r=1.5 | | RGG r=1 | | Ring | |
|---|---|---|---|---|---|---|---|---|---|---|---|---|
|  | Avg | # Comm. | Avg | # Comm. | Avg | # Comm. | Avg | # Comm. | Avg | # Comm. | Avg | # Comm. |
| VFL | 0.430 | 10.6 | 0.406 | 7.4 | 0.407 | 5.2 | 0.375 | 3.5 | 0.386 | 2.0 | 0.385 | 1.4 |
| 4-MVFL | 0.591 | 42 | 0.572 | 29 | 0.555 | 20.4 | 0.517 | 14.8 | 0.488 | 7.98 | 0.485 | 5.6 |
| 4-MVFL-G2 | 0.687 | 126 | 0.661 | 87 | 0.623 | 61.2 | 0.566 | 44.8 | 0.491 | 23.94 | 0.484 | 16.8 |
| MVFL | 0.594 | 168.5 | 0.581 | 114.9 | 0.558 | 80.8 | 0.528 | 58.7 | 0.503 | 33.5 | 0.507 | 22.7 |
| MVFL-G4 | 0.732 | 836.2 | 0.728 | 572.1 | 0.721 | 407.2 | 0.689 | 293.9 | 0.62 | 168.2 | 0.558 | 113.4 |

**Best, Worst and Select Any Metrics**    To better understand our methods, particularly the gossip aggregations, we show the results for all four metrics on 3 datasets for 50% communication fault rate on a complete network in Table 3. In the Appendix, Table 3 also includes 30% communication fault rate. The dropout rate during training for PD-VFL and the communication dropout rate for CD-MVFL is the same at 30%. The trends support our theoretic analysis and support the idea that gossip improves ML performance and reduces client variablity as seen by the gap between the Best and Worst oracle metrics. Furthermore, the improved performance of communication-wise dropout improves the performance of each client individually. This enables CD-MVFL-G4 to match or significantly outperform all other approaches across the two fault rates. Finally, we note that the Any

Rand metric is the hardest because some devices may fail to communicate and thus the prediction is random.

Table 3: Best models for $50\%$ *complete-communication* test fault rates within 1 standard deviation are bolded. More detailed results with standard deviation are shown in the Appendix.

| | | MNIST | | | | SCMNIST | | | | CIFAR10 | | | |
| | | Active | | | Any | Active | | | Any | Active | | | Any |
| | | Worst | Rand | Best | Rand | Worst | Rand | Best | Rand | Worst | Rand | Best | Rand |
|---|---|---|---|---|---|---|---|---|---|---|---|---|---|
| Test Fault Rate = 0.5 | VFL | nan | 0.294 | nan | nan | nan | 0.258 | nan | nan | nan | 0.181 | nan | nan |
| | PD-VFL | nan | 0.488 | nan | nan | nan | 0.424 | nan | nan | nan | 0.263 | nan | nan |
| | 4-MVFL-G2 | 0.564 | 0.612 | 0.721 | 0.423 | 0.466 | 0.519 | 0.620 | 0.368 | 0.251 | 0.303 | 0.387 | 0.228 |
| | MVFL | 0.042 | 0.518 | 0.966 | 0.313 | 0.035 | 0.465 | **0.925** | 0.280 | 0.007 | 0.264 | **0.762** | 0.182 |
| | MVFL-G4 | 0.843 | 0.847 | 0.851 | 0.474 | 0.676 | 0.680 | 0.684 | 0.389 | 0.402 | 0.401 | 0.413 | 0.252 |
| | CD-4-MVFL-G2 | 0.863 | 0.852 | 0.923 | **0.581** | 0.693 | 0.691 | 0.763 | **0.482** | 0.392 | 0.422 | 0.499 | 0.305 |
| | CD-MVFL-G4 | **0.974** | **0.975** | **0.976** | 0.538 | **0.785** | **0.786** | 0.787 | 0.443 | **0.501** | **0.504** | 0.507 | **0.301** |

## 5 CONCLUSION AND DISCUSSION

In this paper, we carefully defined DN-VFL, proposed and theoretically analyzed MAGS as a method for DN-VFL, developed a testbed, and evaluated and compared performance across various fault models and datasets. Simulated faults via dropout increase the robustness of MAGS to distribution shifts. Multiple VFL allows MAGS to avoid catastrophic faults since any device (including active parties) can fail. Gossiping outputs at inference time implicitly ensembles the predictions for neighboring devices that theoretically improves the robustness and reduces the variance. Our work lays the foundation for DN-VFL, opening up many directions for future research, such as handling heterogeneous devices or models, exploring new architectures, and considering different fault models.

Furthermore, we emphasize that our focus was on the machine learning aspects of decentralized VFL robustness. Our level of analysis (e.g., communication cost and simple simulation of devices) is comparable to that of many other studies in HFL (McMahan et al., 2017; Wang et al., 2022b; Crawshaw et al., 2024) and VFL (Li et al., 2023; Castiglia et al., 2023; Jin et al., 2021; Jiang et al., 2022). While we discuss some system-level aspects (communication bottlenecks and latency) in the appendix, real-world deployment would require more detailed system-level research and represents an important future direction. Additionally, we assume a synchronous communication model and asynchronous models could be a natural extension for future systems in the DN-VFL context. Finally, in the appendix, we compare MAGS to an alternative solution using the fault-tolerant consensus algorithms like Paxos or Raft (Lamport, 2001; Ongaro & Ousterhout, 2014), showing that these are insufficient for the DN-VFL context.

Another interesting topic to explore further is the impact of faults during training. As discussed in the appendix, our method could handle faults in the backward propagation phase. However, a more thorough analysis of backward faults—particularly when a high fault rate is expected during training—remains an open issue beyond the scope of this work, where we focus on near-catastrophic faults at *inference*. If very high fault rates were introduced during training, synchronized backpropagation may break down. We hypothesize that fully synchronized backpropagation training may not be ideal in such scenarios. Therefore, localized learning approaches could be beneficial. Methods such as Forward-Forward algorithms (Hinton, 2022), dual propagation (Høier et al., 2023), and other localized learning techniques (Detorakis et al., 2018; Movellan, 1991; Czarnecki et al., 2017; Belilovsky et al., 2019; 2020) could be explored to enhance robustness during training. We leave a more detailed investigation of training with extreme faults to future work. Finally, while we focused on the robustness aspects, in practice, more advanced architectures such as CNN-based or transformer-based could be leveraged for better absolute performance. We believe our MAGS techniques could be easily adapted to different architectures and thus provide an orthogonal contribution compared to architecture design.

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

# Appendix

## Table of Contents

## A  ILLUSTRATION OF METHODS AND FAULTS

### A.1  ILLUSTRATION OF MAGS PROPOSED TECHNIQUES

In Figure 3 we provide a visual representation of the techniques used in MAGS. Furthermore, we provide a representation of VFL in Figure 3 and show how each of the techniques build upon the VFL setup.

### A.2  DEVICE AND COMMUNICATION FAULT VISUALIZATION

We show in Figure 4 the visual representation of communication and device fault under the MVFL method. Although, we present the scenario for only one method, by extension the visualization is similar for VFL, DMVFL and the gossip variants.

### A.3  PARTY-WISE DROPOUT AND COMMUNICATION-WISE DROPOUT MVFL

In Section 3 of the main paper, we presented the Party-wise and communication-wise Dropout method for MVFL. In Figure 5 we represent CD-MVFL and PD-MVFL for a group of 3 devices.

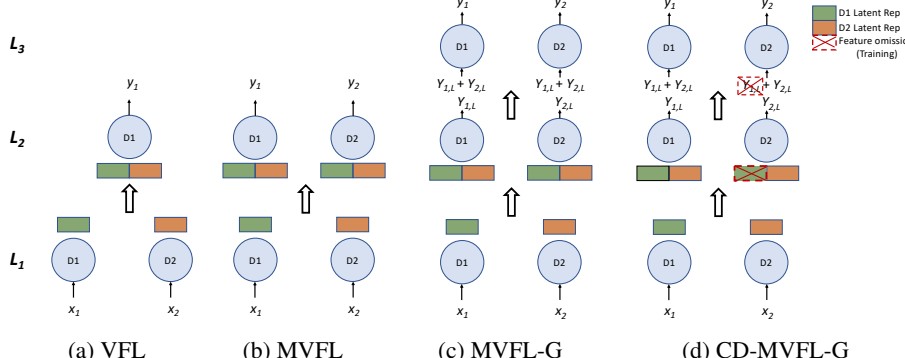

(a) VFL      (b) MVFL      (c) MVFL-G      (d) CD-MVFL-G

Figure 3: VFL and the proposed enhancements are illustrated for a network of two fully connected devices, D1 and D2. (a)VFL setup with D1 acting as a client as well as an aggregator. The input to the devices at the first layer $L_1$ are $x_1$ and $x_2$ and output are the latent representations. The input to the server on the second layer $L_2$ is the concatenated latent representation and the output is the prediction $y_1$ (b) MVFL arrangement has both the devices acting as data aggregators aside from being clients. (c) MVFL-G is an extension of MVFL wherein the output log probabilities $(Y_{i,L})$ from each device are averaged before being used for final prediction. (d) CD-MVFL-G is a variant of MVFL-G where during the training phase, representation from D2 to D1 is not communicated by design. CD-MVFL-G is the method that we propose for DN-VFL

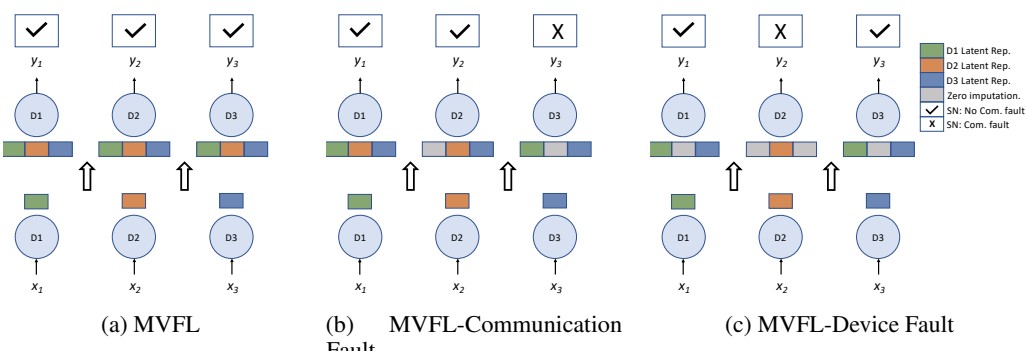

(a) MVFL      (b)    MVFL-Communication Fault      (c) MVFL-Device Fault

Figure 4: Illustration of communication and device faults for a 3 device network for the MVFL method. (a) Fully connected MVFL setup. The check mark indicates that there is no fault in the final communication between device and special node (SN) as defined in Section 3 of the main paper (b) Representation with communication faults. In this example communication from D1 to D2 and D2 to D3 is faulted. To account for the missing values, we do zero imputation. $X$ indicates that the communication between D3 and SN is faulted. Hence, the output at SN will be a class selected with uniform probability among all the classes (c) In device faults, the faulted device do not communicate with any other devices and missing values are accounted for by zero imputation. In this example, D2 is assumed to be faulted, hence the information from D2 is not passed to D1 or D3 and it does not produce an output. The output at SN for $D_2$ will be a class selected with uniform probability among all the classes

## B   PROOFS

### B.1   PROOF OF PROPOSITION 1

Before we prove the proposition, we will prove the following lemma about the conditional probability of a client being selected given a known active set size.

**Lemma 4** (Conditional Client Selection Probability). *The conditional probability of selecting each data aggregator $k \in \mathcal{K} \cup \{\emptyset\}$ (aggregators plus the possible fake aggregator $k = \emptyset$) given a current*

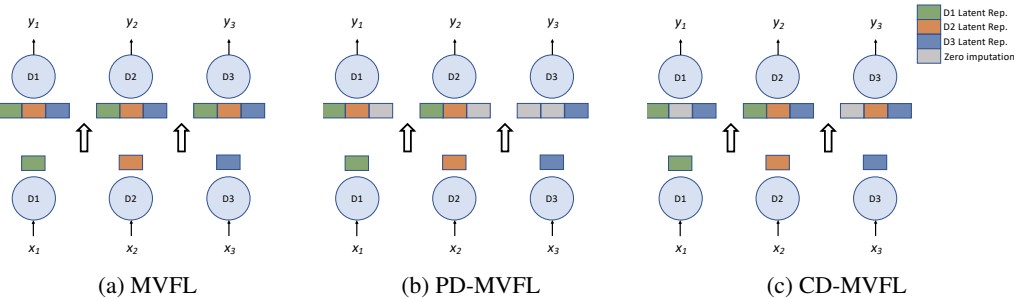

(a) MVFL $\qquad$ (b) PD-MVFL $\qquad$ (c) CD-MVFL

Figure 5: Illustration of Party wise and Communication wise dropout for a 3 device network for the MVFL method. (a) Fully connected MVFL setup. (b) For Party wise Dropout (PD), during training if D3 is dropped then none of the devices gets representations from D3 and the missing values are imputed by zeros. (c) In communication wise Dropout (CD) certain representations are omitted during training. In this example representations from from D2 to D1 and D1 to D3 are omitted by design during the training.

*active set size $|\mathcal{A}|$ is as follows:*

$$p(S = k \,\big|\, |\mathcal{A}|) = \begin{cases} \frac{1}{K}, & \textit{if } |\mathcal{A}| > 0 \textit{ and } k \in \{1, \ldots, K\} \\ 1, & \textit{if } |\mathcal{A}| = 0 \textit{ and } k = \emptyset \\ 0, & \textit{otherwise} \end{cases} . \tag{2}$$

*Proof of Lemma 4.* Let $S$ denote the final selected index. Let $A_k$ denote whether $k$ is in the active set (assuming device or communication fault models). First, we derive the probability of selection for a specific active set size $b$. We notice that $p(S = \emptyset \,\big|\, |\mathcal{A}| = 0) = 1$ (i.e., the fake client is selected if there are no active real clients) and $p(S = \emptyset \,\big|\, |\mathcal{A}| > 0) = 0$ (i.e., the fake client is never selected if there is at least one active client. Similarly, $p(S = k \,\big|\, |\mathcal{A}| = 0) = 0, \forall c$ because there are no active clients. The last remaining case is when $|\mathcal{A}| > 0$ and $b \neq \emptyset$, i.e., $p(S = b \,\big|\, |\mathcal{A}| > 0), \forall b \neq \emptyset$, which we derive as $\frac{1}{K}$ below:

$$p(S = k \,\big|\, |\mathcal{A}| = b) \tag{3}$$

$$= \sum_{a=0}^{1} p(S = k, A_k = a \,\big|\, |\mathcal{A}| = b) \tag{4}$$

$$= \sum_{a=0}^{1} p(A_k = a \,\big|\, |\mathcal{A}| = b) p(S = k \,\big|\, A_k = a, |\mathcal{A}| = b) \tag{5}$$

$$= p(A_k = 1 \,\big|\, |\mathcal{A}| = b) p(S = k \,\big|\, A_k = 1, |\mathcal{A}| = b) + p(A_k = 0 \,\big|\, |\mathcal{A}| = b) p(S = k \,\big|\, A_k = 0, |\mathcal{A}| = b) \tag{6}$$

$$= p(A_k = 1 \,\big|\, |\mathcal{A}| = b) p(S = k \,\big|\, A_k = 1, |\mathcal{A}| = b) + p(A_k = 0 \,\big|\, |\mathcal{A}| = b) \cdot 0 \tag{7}$$

$$= p(A_k = 1 \,\big|\, |\mathcal{A}| = b) p(S = k \,\big|\, A_k = 1, |\mathcal{A}| = b) \tag{8}$$

$$= \left(\frac{b}{K}\right) \left(\frac{1}{b}\right) \tag{9}$$

$$= \frac{1}{K} . \tag{10}$$

where equation 4 is by marginalization of joint distribution, equation 7 is by noticing that if the device is not active, then it will not be selected, and equation 9 is by the uniform distribution for Select Active $h$ and by noticing that

$$p(A_k = 1 \,\big|\, |\mathcal{A}| = b) = \frac{\text{Num. subsets of size } b \text{ with } c \text{ in them}}{\text{Num. subsets of size } b} = \frac{\binom{K-1}{b-1}}{\binom{K}{b}} = \frac{K}{b} \tag{11}$$

where the numerator can be thought of as finding all possible subsets of size $b-1$ from $K-1$ clients (where client $c$ has been removed) and then adding client $c$ to get a subset of size $b$.

Putting this altogether we arrive at the following result for the probability of selection given various sizes of the active set:

$$p(S = k \,|\, |\mathcal{A}| ) = \begin{cases} \frac{1}{K}, & \text{if } |\mathcal{A}| > 0, c \in \{1, \ldots, K\} \\ 1, & \text{if } |\mathcal{A}| = 0, c = \emptyset \\ 0, & \text{otherwise} \end{cases} \tag{12}$$

$\square$

Given this lemma, we now give the proof of the proposition.

*Proof of Proposition 1.* Let $S \in \{\emptyset, 1, 2, \ldots, K\}$ denote a random variable that is the index of the final client prediction selected based on $h$, where $\emptyset$ denotes a fake client that represents the case where a non-active client is selected (which could happen in Select Any Client $h$ or if no clients are active for Select Active Client $h$). The output of this fake client is equivalent to the marginal probability of $Y$ since the external client would know nothing about the input and would be as good as random guessing. Furthermore, let $\Psi_S$ denote the $S$-th client's prediction. Given this notation, we can expand the risk in terms of $S$ instead of $h$:

$$R_h(\theta; \mathcal{G}(t)) \tag{13}$$

$$= \mathbb{E}_{\boldsymbol{x}, y, \mathcal{G}(t), h}[\ell_h(\Psi(\boldsymbol{x}; \theta, \mathcal{G}(t)), y)] \tag{14}$$

$$= \mathbb{E}_S[\mathbb{E}_{\boldsymbol{x}, y, \mathcal{G}(t)|S}[\ell(\Psi_S(\boldsymbol{x}; \theta, \mathcal{G}(t)), y)]] \tag{15}$$

$$= \Pr(S \neq \emptyset)\mathbb{E}_{\boldsymbol{x}, y, \mathcal{G}(t)|S \neq \emptyset}[\ell(\Psi_S(\boldsymbol{x}; \theta, \mathcal{G}(t)), y)] + \Pr(S = \emptyset)\mathbb{E}_{\boldsymbol{x}, y, \mathcal{G}(t)|S = \emptyset}[\ell(\Psi_S(\boldsymbol{x}; \theta, \mathcal{G}(t)), y)] \tag{16}$$

$$= \Pr(S \neq \emptyset)\mathbb{E}_{\boldsymbol{x}, y, \mathcal{G}(t)|S \neq \emptyset}[\ell(\Psi_S(\boldsymbol{x}; \theta, \mathcal{G}(t)), y)] + \Pr(S = \emptyset)R(\theta; \mathcal{G}_{\text{empty}}) \tag{17}$$

$$= (1 - r^K)\mathbb{E}_{\boldsymbol{x}, y, \mathcal{G}(t)|S \neq \emptyset}[\ell(\Psi_S(\boldsymbol{x}; \theta, \mathcal{G}(t)), y)] + r^K R(\theta; \mathcal{G}_{\text{empty}}) \tag{18}$$

$$\tag{19}$$

where the last term is by noticing that the probability of the fake one being chosen is equivalent to $|\mathcal{A}| = 0$ and thus all devices fail which would have a probability of $r^K$. We now decompose the second term in terms of clean risk:

$$\mathbb{E}_{\boldsymbol{x}, y, \mathcal{G}(t)|S \neq \emptyset}[\ell(\Psi_S(\boldsymbol{x}; \theta, \mathcal{G}(t)), y)] \tag{20}$$

$$= \mathbb{E}_{S|S \neq \emptyset}[\mathbb{E}_{\boldsymbol{x}, y, \mathcal{G}(t)|S}[\ell(\Psi_S(\boldsymbol{x}; \theta, \mathcal{G}(t)), y)]] \tag{21}$$

$$= \mathbb{E}_{S|\|\mathcal{A}\|>0}[\mathbb{E}_{\boldsymbol{x}, y, \mathcal{G}(t)|S}[\ell(\Psi_S(\boldsymbol{x}; \theta, \mathcal{G}(t)), y)]] \tag{22}$$

$$= \sum_k p(S = k|\|\mathcal{A}\| > 0)\mathbb{E}_{\boldsymbol{x}, y, \mathcal{G}(t)|S}[\ell(\Psi_S(\boldsymbol{x}; \theta, \mathcal{G}(t)), y)] \tag{23}$$

$$= \sum_k \frac{1}{K}\mathbb{E}_{\boldsymbol{x}, y, \mathcal{G}(t)|S}[\ell(\Psi_S(\boldsymbol{x}; \theta, \mathcal{G}(t)), y)] \tag{24}$$

$$= \sum_k \frac{1}{K}R_{h_k}(\theta; \mathcal{G}(t)) \tag{25}$$

$$\geq \sum_k \frac{1}{K}R_{h_k}(\theta; \mathcal{G}_{\text{clean}}) \tag{26}$$

$$= \sum_k \frac{1}{K}R_{h_k}(\theta; \mathcal{G}_{\text{clean}}) \tag{27}$$

$$= R_h(\theta; \mathcal{G}_{\text{clean}}), \tag{28}$$

where equation 24 is by Lemma 4, the inequality is due to our assumption that risk on a faulty graph is less than the risk on a clean graph, and the last line is by definition of the clean risk where $h$ is $h_{\text{active}}$. Combining the results, we have the final result:

$$R_h(\theta; \mathcal{G}(t)) = (1 - r^K)R_h(\theta; \mathcal{G}_{\text{clean}}) + r^K R(\theta; \mathcal{G}_{\text{empty}}). \tag{29}$$

$\square$

## B.2 Proof of Proposition 2

*Proof.* We first note that using a geometric average of probabilities (implemented using log probabilities for stability) satisfies the conditions in the Generalised Ambiguity Decomposition Wood et al. (2023, Proposition 3) for the ensemble combiner. (Similarly, if the problem was regression, we could use the squared loss with an arithmetic mean ensemble combiner for gossip.) As a reminder, let $\Psi_k$ denote the models that output probabilities of each class and let the ensemble model be denoted as $\Psi_{\text{ens}}$ where $\Psi_k^{\text{ens}}(\boldsymbol{x}; \theta, \mathcal{G}(t)) \triangleq Z^{-1} \exp(\sum_{k'=1}^K \Psi_{k'}(\boldsymbol{x}; \theta, \mathcal{G}(t))), \forall k \in \mathcal{K}$, where $Z$ is the normalizing constant to ensure the final output is a probability vector. Furthermore, let $\Psi_h(\boldsymbol{x}; \theta, \mathcal{G}(t)) \triangleq h(\Psi(\boldsymbol{x}; \theta, \mathcal{G}(t)), \mathcal{G}(T))$, i.e., it is merely the postprocessing of the original $\Psi$ function with $h$. This allows us to interchange the $h$ between the loss function and a modified $\Psi$, i.e., $\ell_h(\Psi(\boldsymbol{x}; \theta, \mathcal{G}(t)), y) = \ell(\Psi_h(\boldsymbol{x}; \theta, \mathcal{G}(t)), y)$. Similarly, with a slight abuse of notation, if $\Psi$ is on both sides of the loss function, we will apply $h$ to both inputs before passing to the loss function, i.e., $\ell_h(\Psi_1(\boldsymbol{x}; \theta, \mathcal{G}(t)), \Psi_2(\boldsymbol{x}; \theta, \mathcal{G}(t))) = \ell(\Psi_{h,1}(\boldsymbol{x}; \theta, \mathcal{G}(t)), \Psi_{h,2}(\boldsymbol{x}; \theta, \mathcal{G}(t)))$. Given this, assuming that $h = h_{\text{active}}$, we can decompose the risk as follows:

$$R_h^{\text{ens}}(\theta; \mathcal{G}(t)) \tag{30}$$

$$= \mathbb{E}_{\boldsymbol{x}, y, \mathcal{G}(t), h}[\ell_h(\Psi^{\text{ens}}(\boldsymbol{x}; \theta, \mathcal{G}(t)), y)] \tag{31}$$

$$= \mathbb{E}_{\boldsymbol{x}, y, \mathcal{G}(t), h}[\ell(\Psi_h^{\text{ens}}(\boldsymbol{x}; \theta, \mathcal{G}(t)), y)] \tag{32}$$

$$= \mathbb{E}_{\boldsymbol{x}, y, \mathcal{G}(t), h}[\tfrac{1}{K} \sum_{k=1}^K \ell(\Psi_{h,k}(\boldsymbol{x}; \theta, \mathcal{G}(t)), y) - \tfrac{1}{K} \sum_{k=1}^K \ell(\Psi_{h,k}(\boldsymbol{x}; \theta, \mathcal{G}(t)), \Psi_h^{\text{ens}}(\boldsymbol{x}; \theta, \mathcal{G}(t)))] \tag{33}$$

$$= \mathbb{E}_{\boldsymbol{x}, y, \mathcal{G}(t), h}[\tfrac{1}{K} \sum_{k=1}^K \ell_h(\Psi_k(\boldsymbol{x}; \theta, \mathcal{G}(t)), y) - \tfrac{1}{K} \sum_{k=1}^K \ell_h(\Psi_k(\boldsymbol{x}; \theta, \mathcal{G}(t)), \Psi^{\text{ens}}(\boldsymbol{x}; \theta, \mathcal{G}(t)))] \tag{34}$$

$$= R_h(\theta; \mathcal{G}(t)) - \mathbb{E}_{\boldsymbol{x}, \mathcal{G}(t), h}[\tfrac{1}{K} \sum_{k=1}^K \ell_h(\Psi_k(\boldsymbol{x}; \theta, \mathcal{G}(t)), \Psi^{\text{ens}}(\boldsymbol{x}; \theta, \mathcal{G}(t)))] \tag{35}$$

$$\leq R_h(\theta; \mathcal{G}(t)) \,. \tag{36}$$

where the first equals is by definition, the second is by pushing the $h$ function into $\Psi$ so that we have the raw loss function $\ell$, the third equals is by Wood et al. (2023, Proposition 3), the fourth is by pulling the $h$ function back out into the loss function with a slight abuse of notation where the RHS term the $h$ function is applied to both arguments before passing to the original loss function, the fifth is by noticing that the non-ensemble risk is equal to the average risk of each aggregator-specific model, and the last inequality is by noticing that hte loss function is always non-negative. □

## C Handling Backward Pass for Training with Faults

In the forward pass, faulted messages can be merely treated as dropout. We discuss handling of faults in the backward pass in the next paragraphs.

In the backward pass, the gradients of the classifier head are computed locally on each client, unaffected by faults. However, calculating the gradients for the feature encoders requires an additional communication round to send gradients back to each client, as in standard VFL. Only the gradients for non-faulted or non-dropped messages need to be sent, as dropped messages are treated as zeros in the forward pass. Faulty gradient messages can similarly be imputed with zeros as in forward pass dropout since a dropped forward message is functionally equivalent to a dropped gradient message from the perspective of the feature encoder.

In practice, we assume that the gradient communication round will typically succeed, as we expect the device network to be stable during normal operation. This assumption is reasonable for training, given that the fault pattern is unlikely to change significantly over the short time required to process a batch of data, provided there are no catastrophic events like extreme weather disruptions. Note that the fault pattern may vary between batches, but it only needs to remain stable for two communication rounds per batch.

## D Deep MVFL

In Section 3 of the main paper, we presented a few innovations that introduces redundancy in the system. Extending the MVFL setup, we propose another variant, Deep MVFL (DMVFL), which

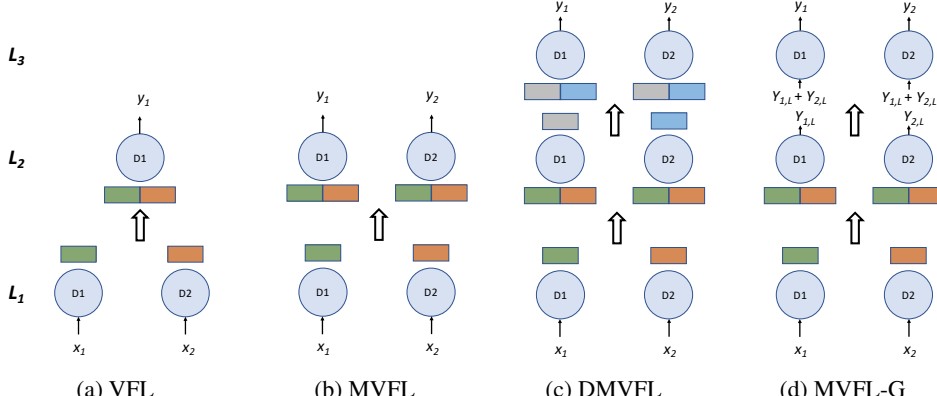

(a) VFL          (b) MVFL          (c) DMVFL          (d) MVFL-G

Figure 6: VFL as a baseline and the proposed innovations are illustrated for a network of two fully connected devices, D1 and D2. (a)VFL setup with D1 acting as a client as well as the aggregating server. The input to the devices at the first layer $L_1$ are $x_1$ and $x_2$ and output are the latent representations. The input to the server on the second layer $L_2$ is the concatenated latent representation and the output is the prediction $y_1$ (b) MVFL arrangement has both the devices acting as servers aside from being clients. (c) DMVFL has a similar arrangement as MVFL, expect that there is an additional layer of processing, $L_3$, that has the concatenated features from the previous layer as an input and the output are the predictions. (d) MVFL-G is an extension of MVFL wherein the output log probabilities ($Y_{i,L}$) from each device are averaged before being used for final prediction.

stacks MVFL models on top of each other and necessitates multiple rounds of communication between devices for each input. We believe that multiple communication rounds and deeper processing could lead to more robustness on dynamic networks. Comparing Figure 6 (b) and (c), the setup is same till $L_2$ but in DMVFL, there is an additional round of communication in $L_3$ following which the predictions are made. In Figure 6 (c) we have illustrated DMVFL with just one additional round of communication over MVFL and hence the depth of DMVFL is 1. Nonetheless, the depth in DMVFL need not be restricted to 1 and is a hyperparameter. Furthermore, to guarantee a fair comparison between DMVFL and MVFL, it was ensured that the number of parameters for both these setups be the same.

In DMVFL the redundancy is over depth, However, based on our experiments we did not observe a significant performance gain and hence did not present it in the main paper.

## E  FURTHER DISCUSSION AND LIMITATIONS

**Designing for Extreme Fault Rates**    We argue that designing for extreme fault rates of 50% is a valid approach even if 50% fault rates are not common in current edge networks. We provide at least two main arguments below.

First, our problem setup is inspired by other robust methods that continue to operate in very poor conditions even if these extreme conditions differs from the average case. For example, the fault-tolerant distribute consensus algorithm Paxos (Lamport, 2001) is safe even with arbitrarily bad failures. However, in practice, Paxos may be used even when the failure rate is very low. Similarly, internet protocols were designed to operate even under arbitrary network faults (known as "survivability" in early packet switching papers (Baran, 1964)) even though the average case has a small fault rate. Thus, even if 50% fault rate is uncommon, it is important to design systems that will operate even under the uncommon events.

Second, we expect that 50% fault rate is reasonable in three scenarios: harsh environmental conditions, cheap and unreliable sensors, and rare disaster-like events. In harsh environments like deep-sea or remote wilderness regions, a fault rate of 50% might be reasonable, especially over a long period of time. Similarly, if one uses a large number of very cheap but unreliable sensors, the use case of 50% fault rate could be quite reasonable. Finally, we expect that a 50% fault rate could be reasonable in disaster scenarios such as floods or hurricanes where conditions are significantly worse than normal.

**Privacy:** In our work we did not focus on privacy since we assumed a trusted , though unreliable network of devices. Privacy is a natural question in untrusted networks. Some work on VFL privacy using homeomorphic encryption or blockcahin-based approach (Li et al. (2023); Tran et al. (2024)) could be extended to our framework as well. We want to say that our contribution is orthogonal and complementary to advancements in VFL privacy.

**Simplified Comparison of VFL and MAGS in Terms of Latency, Throughput, and Power:** We first compare our MAGS method to standard VFL using simplified models of computation and communication and discuss more practical considerations in the next section. Specifically, let $C_1, C_2$ and $M$ denote the maximum time for each device to compute its features, compute a prediction given messages, and exchange messages, respectively. For VFL, the latency would be $C_1 + M + C_2$. For MAGS, the latency would be $C_1 + M + C_2 + G \cdot M$, where $G$ is the number of gossip rounds. If there is no gossip, then MAGS is equivalent to VFL latency. However, as shown in our results, gossip increases the robustness by about 3%-4% compared to no gossiping. Thus, gossip produces a small tradeoff between robustness and latency.

When analyzing throughput, we assume that each device can handle a batch size of $B$ samples for both encoding and prediction. The only difference for throughput between VFL vs MAGS is that MAGS performs extra gossip rounds. We expect these gossip rounds to have negligible impact on throughput as they are only exchanging logit values.

For power, we make the simplifying assumption that power is directly proportional to the number of messages sent by each device. Thus, to analyze average power usage, we point to Table 2 that shows the communication costs of various methods. Here again, we see a tradeoff between using full MVFL or using $k$-MVFL with and without gossip. The best in terms of robustness is full MVFL with gossip rounds but the power consumption would be high. In practice, our method provides different ways to adjust the communication depending on the desired robustness versus power tradeoff. On the other hand, standard VFL has no way of adjusting the communication cost other than simply randomly dropping communications, which may significantly hinder the performance.

**Systems-Level Aspects for Real-World Deployment:** While our paper focuses on the ML challenge of robustness and thus we use simplified assumptions common in the the VFL literature (Li et al., 2023; Castiglia et al., 2023; Jin et al., 2021; Jiang et al., 2022). Nonetheless, practical real-world deployment would likely require analyzing and optimizing for systems-level concerns. While these systems-level aspects for deployment on real networked devices are out of scope for this current ML-focused paper, we discuss some of these aspects for completeness.

- *Device heterogeneity*: Although we assume that all devices have similar capabilities and can thus serve as aggregators, in practice, the devices may have high heterogeneity where some have much higher capacity for computation and communication. A natural extension of our work is to optimize the tradeoff between performance and latency or power as done in Shao et al. (2024); Wang et al. (2022a). Additionally, with heterogeneous devices, more careful design of different model sizes depending on each device's memory capacity could be explored as inLu et al. (2021); Ahmed et al. (2021).

- *Bandwidth*: The experiments in the main paper were simulated with unlimited bandwidth, i.e., no communication bottlenecks. However, it is a salient practical consideration. It may not be possible to perform an all-to-all broadcast over a real wireless network even if all devices have links between them. Instead, devices may need to randomly choose a subset of neighbors to send their messages to. Thankfully, however, MAGS is inherently robust to message losses so devices could choose how often to communicate based on bandwidth or power considerations. Additionally, for real-world deployment, it would be crucial to use careful compression and coding techniques and bandwidth-aware communication to develop variants of MAGS that can accommodate communication bottlenecks.

- *Latency*: While we assume a simplified view of latency in our paper, in practice, some devices will complete their computations faster and there will be some straggler devices. In practice, setting the appropriate cutoff waiting time for messages (i.e., treating delayed messages as faulted) would allow the system to optimize the trade-off between performance and other metrics like latency and power. Again, MAGS can naturally handle this because a delayed message can simply be treated as a faulted message. Additionally, in the future, a natural extension of our work is to develop asynchronous or semi-synchronous variants of

MAGS like Chen et al. (2020); Li et al. (2020), which allow asynchronous updates in the vanilla VFL setting.

**Alternative Approach using Fault-Tolerant Consensus Algorithms**    Instead of direct replication via MVFL, one alternative fault-tolerant approach would be to first run a fault-tolerant consensus algorithm such as Paxos (Lamport, 2001) or Raft (Ongaro & Ousterhout, 2014) and then run standard VFL inference with the elected leader. This could reduce the communication load during distributed inference but would reduce the robustness or increase latency compared to MAGS. For example, Paxos may fail or wait indefinitely for extreme fault rates near 50%.

Additionally, Paxos would increase the latency as consensus would need to be arrived before continuing. On the other hand, MAGS would provide an answer (perhaps degraded but that is expected) with the same latency no matter the percentage of faults even for more extreme faults beyond 50%. Secondly, we point out that accuracy performance with MAGS actually benefits between 2-3% because of the ensembling of multiple devices' predictions. For example, on the grid graph in Figure 2c in the main paper, there is a 2-3% gap between PD-MVFL (which would roughly correspond to PD+Paxos) compared to PD-MVFL-G4 (which has gossip and produces an ensembling effect).

While distributed algorithms such as Paxos and Raft (Lamport, 2001; Ongaro & Ousterhout, 2014) are useful to generate consensus in a system that encounters fault, they do not enable representations that are robust to faults, which is essential for achieving good performance. Thus, they cannot be naively applied for addressing DN-VFL and more carefully constructed method like MAGS is required.

**Distributed Inference Algorithm's Resemblance to GNNs:**    The form of our distributed inference algorithm in the main paper has a superficial resemblance of the computation of graph neural networks (GNN) (Scarselli et al., 2008) but with important semantic and syntactic differences. Semantically, unlike GNN applications whose goal is to predict global, node, or edge properties based on the graph edges, our goal is to do prediction well given *any arbitrary* edge structure. Indeed, the edges in our dynamic network are assumed to be independent of the input and task—rather they are simply constraints based on the network context of the system. Syntactically, our inference algorithm differs from mainstream convolutional GNNs because convolutional GNNs share the parameters across clients (i.e., $\theta_c^{(t)} = \theta^{(t)}$) whereas in our algorithm the parameters at each client are *not shared* across clients (i.e., $\theta_c^{(t)} \neq \theta_{c'}^{(t)}$). Additionally, most GNNs assume the aggregation function $g$ is permutation equivariant such as a sum, product or maximum function. However, we assume $g$ could be any aggregation function. Finally, this definition incorporates the last processing function $h$ that represents the final communication round to an external entity (Main Paper Section 3).

# F   EXPERIMENT DETAILS

## F.1   DATASETS

For the experiments presented in this paper, following are the datasets that were used:

**StarCraftMNIST(SCMNIST):** Contains a total of 70,000 28x28 grayscale images in 10 classes. The data set has 60,000 training and 10,000 testing images. For experiments, all the testing images were used, 48,000 training images were used for training and 12,000 training images were used for validation study.

**MNIST:** Contains a total of 70,000 28x28 grayscale images in 10 classes. The data set has 60,000 training and 10,000 testing images. For experiments, all the testing images were used, 48,000 training images were used for training and 12,000 training images were used for validation study.

**CIFAR-10:** Contains a total of 60,000 32x32 color images in 10 classes, with each class having 6000 images. The data set has 50,000 training and 10,000 testing images. For experiments, all the testing images were used, 40,000 training images were used for training and 10,000 training images were used for validation study.

**CIFAR-100:** Contains a total of 60,000 32x32 color images across 100 classes, with each class having 600 images. The dataset is split into 50,000 training images and 10,000 testing images. For experiments, all the testing images were used, while 40,000 training images were used for training and 10,000 training images were reserved for validation.

**Tiny ImageNet:** Tiny ImageNet consists of 200 classes, each containing 500 64x64 color images for training, 50 images for validation, and 50 images for testing. The dataset includes a total of 100,000 training images, 10,000 validation images, and 10,000 test images. For experiments, all test images were used, and a portion of the training set could be reserved for validation purposes.

## F.2 GRAPH CONSTRUCTION

In the main paper as well as in the Appendix the terms client and devices are used interchangeably. In Section 4 of the main paper, four different graphs were introduced: *Complete,Ring*, *Random Geometric* and *Grid*. To elaborate how these graphs are constructed for a set of 16 clients, we take an example of an image from each of the three datasets and split it up into 16 sections, as illustrated in Figure 7.

For a *Complete* graph, all the devices are connected to the server. For instance, if $D_1$ is selected as the server, then all the other devices $D_i$ for $i = 2, 3, \ldots, 16$ are connected to $D_1$. To construct *Grid* graph, we use compute a *Distance* parameter. For *Grid* graph, *distance* returns true if a selected device lies horizontally or vertically adjacent to a server and only under this circumstance it is connected to the server otherwise it is not. For example, in Figure 7, if $D_3$ is selected as the server, then $D_2$, $D_4$ and $D_7$ are the only devices connected to $D_3$. Another example will be, if $D_{13}$ is selected as the server, then $D_9$ and $D_{14}$ are the only ones connected to the server.

The *Ring* graph is connected by joining all the devices in a sequential order of increasing indices with the last device connected to the first. In our example it will be constructed by joining, $D_1$ with $D_2$, then $D_2$ with $D_3$ and so on and so forth with $D_{16}$ connected back to $D_1$. Finally, *Random Geometric Graph* is constructed by connecting a device to all other devices that fall within a certain radius (r) parameter. Hence a lower value of r denotes a device is connected to less devices compared to a larger value of r.

Irrespective of the base graph, *Grid* or *Complete*, when training or testing faults are applied to the selected base graph, during implementation it is assumed that the graph with incorporated faults stays constant for one entire batch and then the graph is reevaluated for the next batch. In our experiments, the batch-size is taken to be 64.

Furthermore, in Figure 8 we highlight a few examples, to illustrate with MNIST images, why in some cases it is easy to distinguish between images based on partial information and while in other situations, it is not. Thus, device connectivity plays a crucial role in enabling classification tasks.

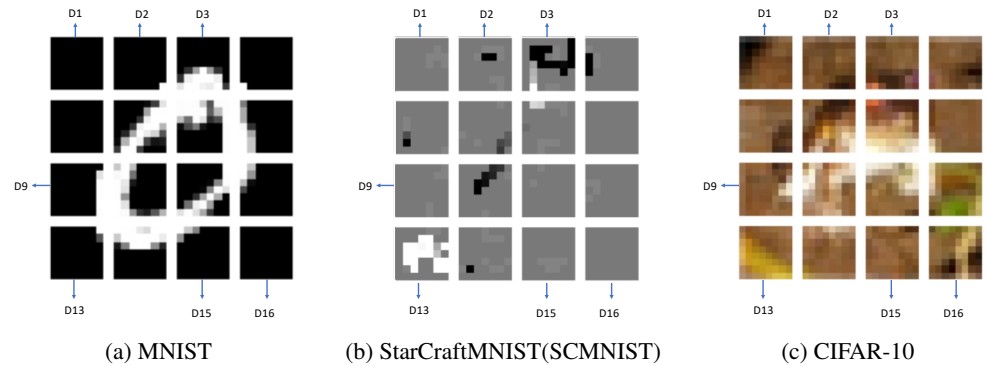

|              (a) MNIST              |    (b) StarCraftMNIST(SCMNIST)    |           (c) CIFAR-10            |

Figure 7: (a)MNIST, (b)SCMNIST, (c)CIFAR-10 Image split into 16 sections. Each section is assigned to a device/client. $D_i$ denotes a device/client

## F.3 TRAINING

For all of our experiments, we train the model for 100 epochs and we always report the result using the model checkpoint with lowest validation loss. We use a batch size of 64 and Adam optimizer with learning rate 0.001 and $(\beta_1, \beta_2) = (0.9, 0.999)$. All experiments are repeated using seed 1,2,...,16.

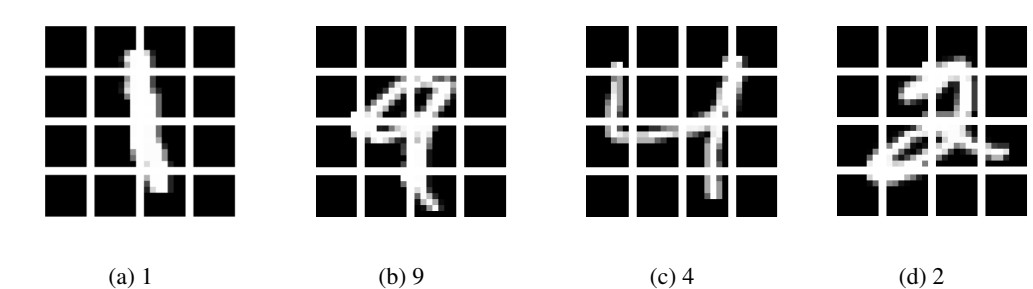

(a) 1            (b) 9            (c) 4            (d) 2

Figure 8: Based on limited information, some images are easy to distinguish from one another, others are not. For instance, based on the information from just bottom half of the devices, it is hard to distinguish between (a) and (b) while differentiating between images (c) and (d) is achievable just based in the bottom half of devices.

For all experiments, we use concatenation as the message passing algorithm where missing values were imputed using zeros (equivalent to dropout).

For 16 devices with all datasets, each device in VFL and MVFL has a model with the following structure: `Linear(49,16),ReLU,Linear(16,4),ReLU,MP,Linear(64,64),ReLU,Linear(64,10)` where `MP` means message passing. For 4 devices with all datasets, each device in VFL and MVFL has a model with the following structure: `Linear(196,64),ReLU,Linear(64,16),ReLU,MP,Linear(64,64),ReLU,Linear(64,10)`. For 49 devices with all datasets, each device in VFL and MVFL has a model with the following structure: `Linear(16,4),ReLU,Linear(4,2),ReLU,MP,Linear(98,98),ReLU,Linear(98,10)`.

For 16 devices with all datasets, each device in DMVFL has a model with the following structure: `Linear(49,16),ReLU,Linear(16,4),ReLU,MP,DeepLayer,Linear(64,10)`. For 4 devices with all datasets, each device in DMVFL has a model with the following structure: `Linear(196,64),ReLU,Linear(16,4),ReLU,MP,DeepLayer,Linear(64,10)`. For 49 devices with all datasets, each device in DMVFL has a model with the following structure: `Linear(16,4),ReLU,Linear(16,4),ReLU,MP,DeepLayer,Linear(98,10)`. `DeepLayer` are composed of a sequence of `MultiLinear(64,16)` based on depth and `MultiLinear(x)=Mean(ReLU(Linear(64,16)(x)),...,ReLU(Linear(64,16)(x)))`. Here we use multiple perceptrons at each layer to make sure the number of parameters between MVFL and DMVFL. For example, for 16 devices and a depth of 2, we use $16/2 = 8$ perceptrons at each layer.

All experiments are performed on a NVIDIA RTX A5000 GPU.

### F.4 HANDLING FAULTS AT TEST TIME

During inference, if a communication or device faults, we impute the missing values with zeros for all methods. Future work could look into other missing value imputation methods that are more effective for the given context.

## G ADDITIONAL EXPERIMENTS

In this section we present some more results from different experiments that we conducted. Like the results in the main paper, we present the Rand test metric over the active set. As we have multiple replicates of an experiment, we take an expectation over the collected Rand metric and this has an averaging effect. Thus, the y axis in the plots of the Appendix are labelled as Test Avg, which is equivalent to Test Active Rand y axis label used in the main paper and these two ways to refer to the metric are used interchangeably.

Furthermore, in our experiments we were initially using gossip both during inference and training. However, we realised that gossip during inference alone is a better approach. Thus, the results in the main paper are presented using gossip only during inference. On the other hand, the experiments

presented in the Appendix use gossip both during inference and training, unless explicitly stated otherwise.

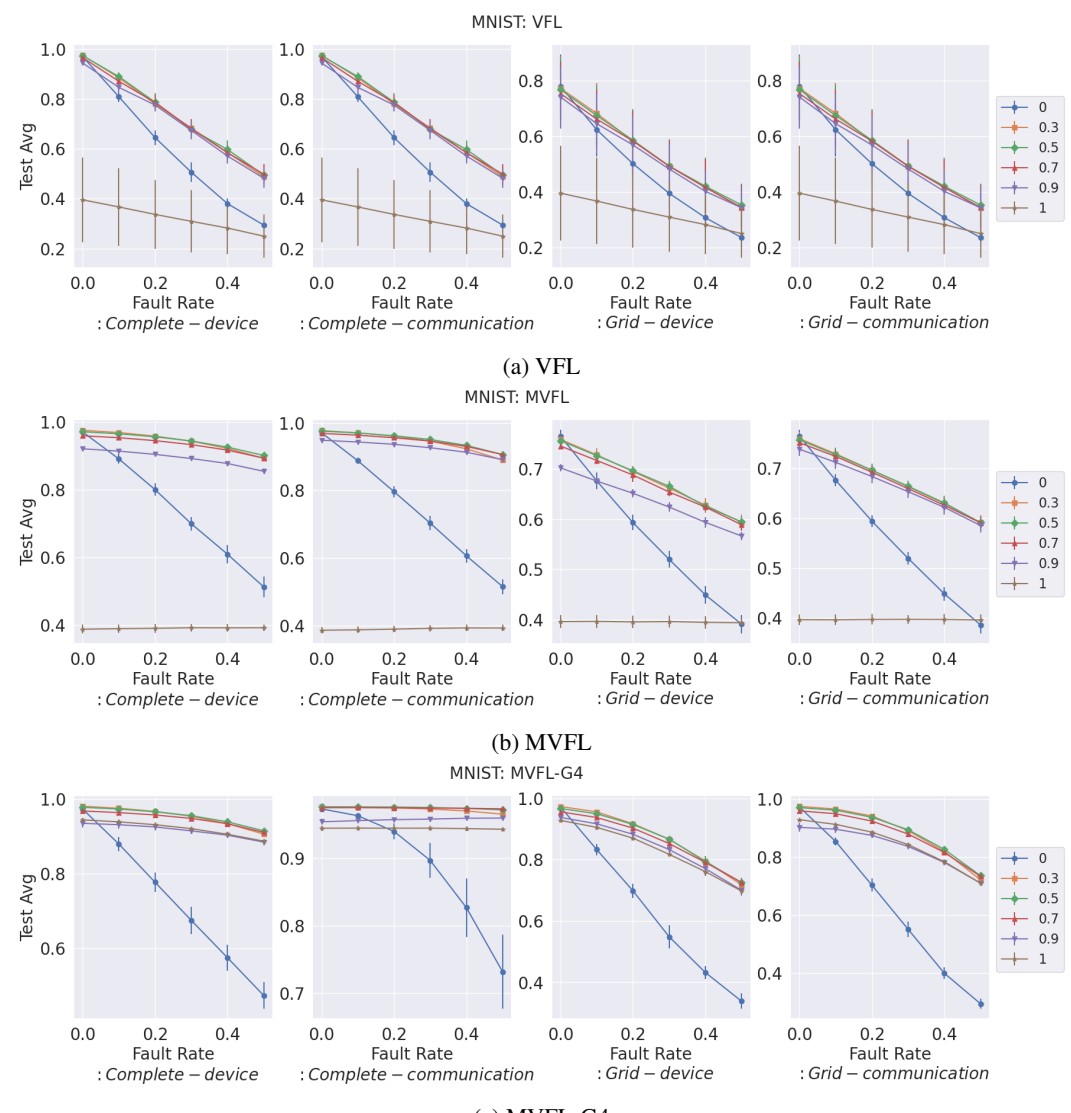

(a) VFL

(b) MVFL

(c) MVFL-G4

Figure 9: Test average accuracy with different dropout rates for MNIST with 16 devices. Across different configurations,training with dropout makes the model robust against test time faults

### G.1 PARTYWISE DROPOUT (PD) AND COMMUNICATION WISE DROPOUT (CD) RATES

In the main paper we presented results with assuming a Dropout rate of 30% for the PD and CD variants. Here we present the effect of different dropout rates on the test time performance under communication and device faulting regime. Figures 9, 10 and 11 show the results for three datasets, MNIST, SCMNIST and CIFAR-10, respectively. Irrespective of communication or device fault scenario, training VFL with an omission rate results in Party wise Dropout. Whereas, for MVFL when studying communication fault having an omission rate results in CD-MVFL model while studying device fault results in PD-MVFL model.

Across the different sets and models it is observed that using CD and PD variants results in improving the performance during test time faults. Furthermore, on observing Figures 9, 10 and 11 (c), it seems that gossiping has a profound impact on the model performance even if the omission rates are

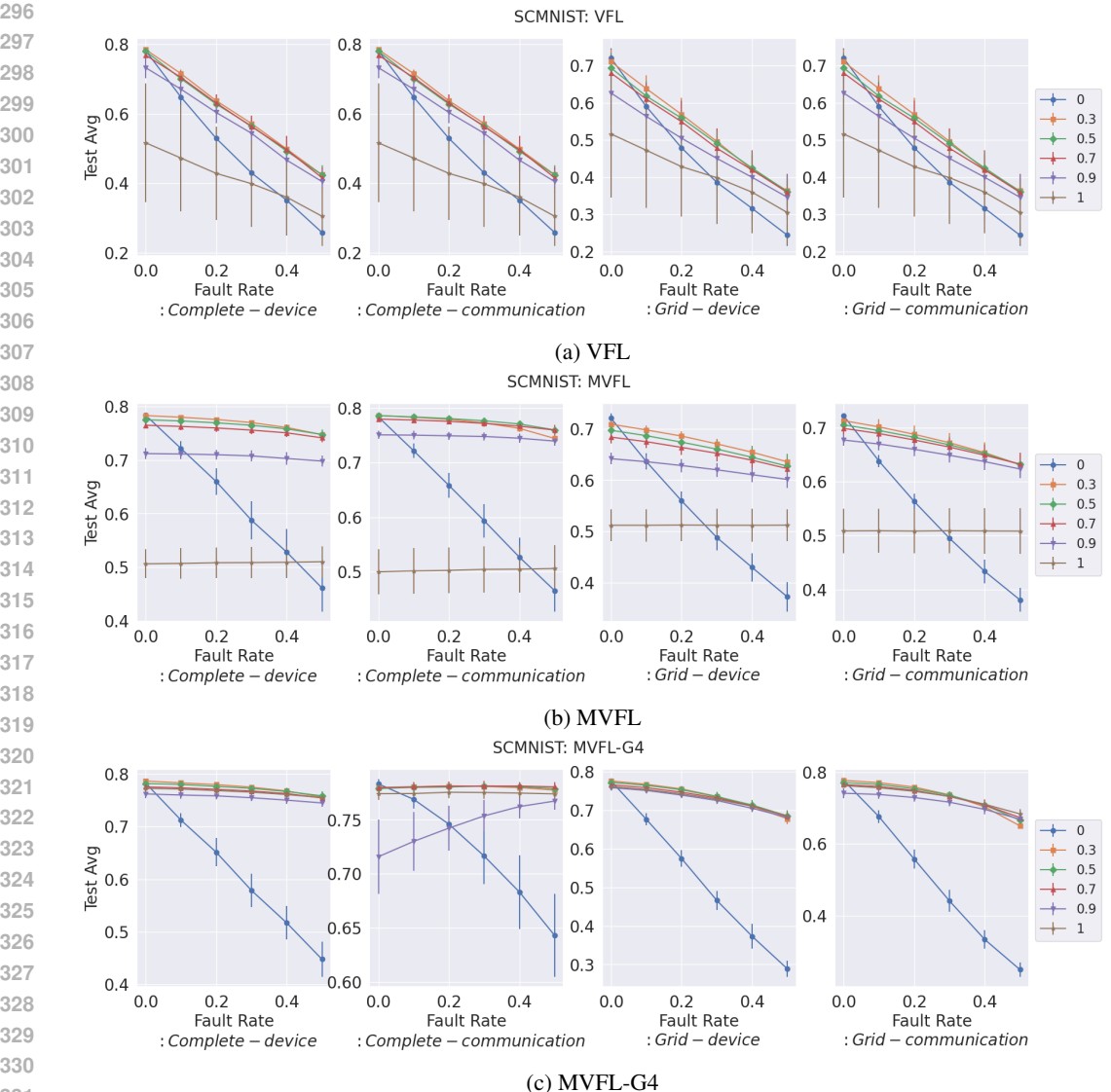

Figure 10: Test average accuracy with different dropout rates for SCMNIST with 16 devices. Across different configurations,training with dropout makes the model robust against test time faults

100% , which means that each device is training it's own local model independently. However, by doing gossip during the testing time, devices are able to reach a consensus that gives the model a performance boost, even during high Dropout rates.

## G.2 EVALUATION FOR A TEMPORAL FAULT MODEL

In Figure 19 we show results with a temporal communication fault model on a complete connected graph. CD- models are trained under a dropout rate of 30%. The temporal fault model uses Markov process to simulate the transition of links or edges in the network between connected and faulty states based on probabilistic rules defined by a transition matrix. The transition matrix is such that when fault rate = 0, the probability of staying in non-faulted state is 1. However for other fault rates, the transition matrix is: [[p, 1-p],[q , 1-q]], where r is the fault rate, q = (1-p)(1-r)/r, and p is fixed at 0.9. p denotes the probability of staying in non-faulted state and q depicts the probability of going from faulted to non-faulted state. Even on a temporally varying graph, MAGS outperforms other baselines.

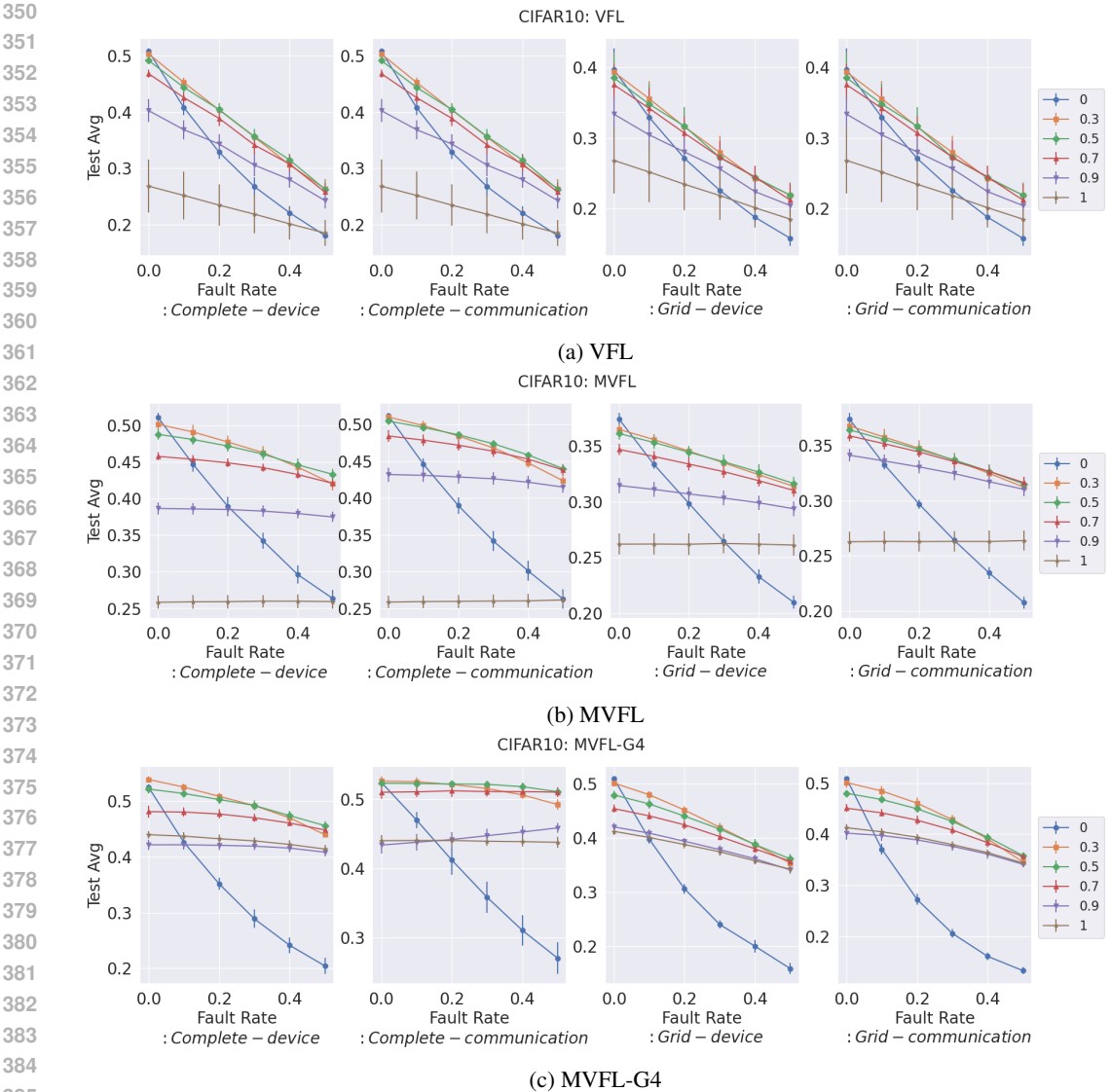

Figure 11: Test average accuracy with different dropout rates for CIFAR10 with 16 devices. Across different configurations,training with dropout makes the model robust against test time faults

## G.3    RESULTS WITH CIFAR100 AND TINY IMAGENET

In Figure 20 we present results with Cifar100 and Tiny ImageNet for a complete graph with communication faults. The trends observed in Figure 20(a) and (b) are similar to what was observed in Figure 2 of the main paper. MVFL and its variants perform better than VFL. The overall performance for Cifar100 and Tiny ImageNet is not comparable to the state of the art classification results as in our experiments we are using only 2 linear layers for classification as in our other experiments to avoid excessive computation. More advanced architectures would be needed to achieve strong classification results. However, our goal is not to compare architectures but to compare the robustness of various approaches. Thus, these results corroborate the findings and trends in our original paper.

## G.4    ABLATION STUDIES

**Choice of number of Aggregators:**    As an initial investigation, we studied the performance of MVFL with different numbers of aggregators for the 16 device grid, complete communication and random geometric graph with a 2.5 radius(r) setting with a 30% communication fault rate during

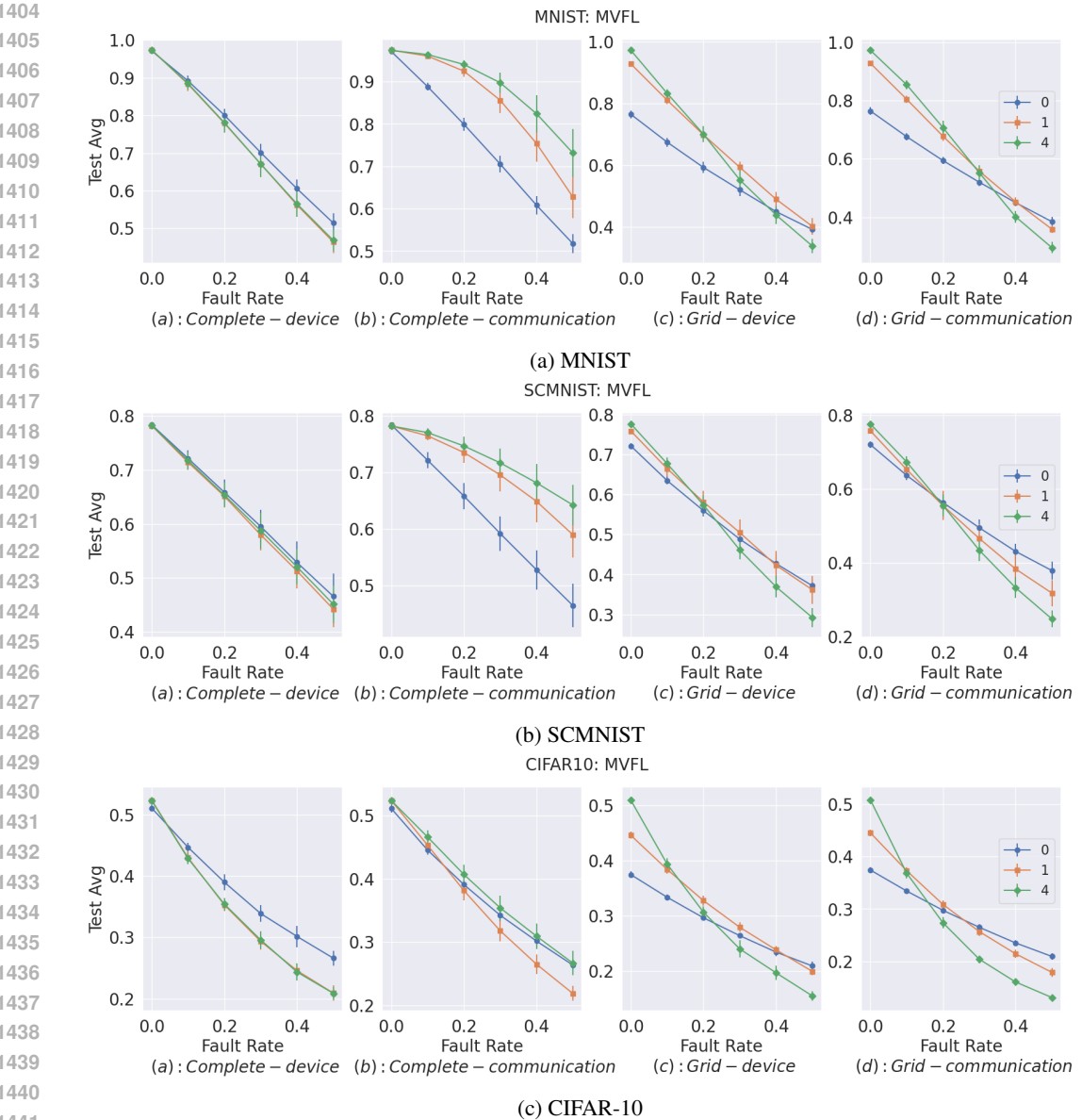

Figure 12: MVFL: Effect of different rounds of Gossip on average performance when evaluated with test time faults for 16 devices

inference. The number of aggregators could be 1 (Standard VFL), 2, 4, 16 (Original MVFL). If the number of aggregators is less than 16, then they were chosen at random with uniform probability. From the Tables 4 to 6 presented below, we see that the major improvement over VFL is achieved by just having 4 devices acting as aggregators. Thus, a major performance boost over VFL can be achieved at a minimal increase in communication cost, and shows that it is not necessary to have all the devices act as aggregators and incur large communication overhead. Following this result, one can infer that gossip variants of setup with few number of data aggregators than MVFL will have a significant less communication overhead than gossip variant of MVFL.

**Effect of number of gossip rounds:** For the gossip (G) variants of MVFL, we are interested in studying the effect the number of gossip rounds has on average performance. In Figure 12 the effect of three different gossip rounds on the average performance for three different datasets is presented. From the plots we observe that irrespective of the method, *Complete-communication* train fault benefits the most with incorporating gossip rounds. Despite *Grid-communication* being

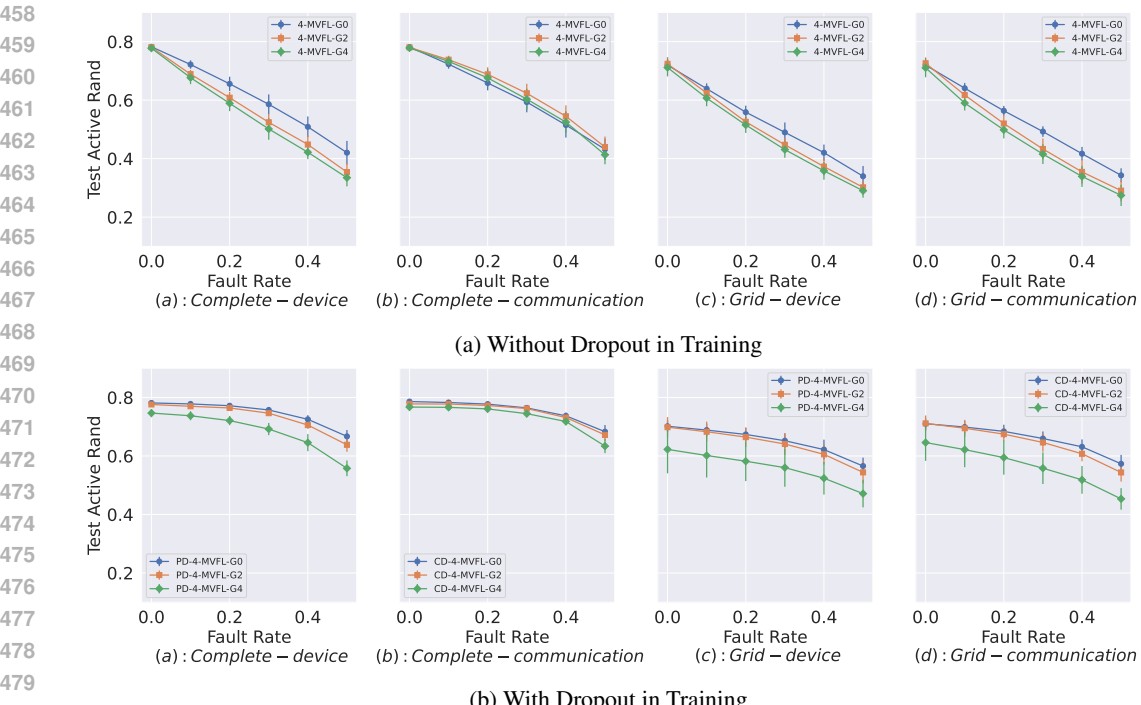

(a) Without Dropout in Training

(b) With Dropout in Training

Figure 13: 4-MVFL: Effect of different rounds of Gossip on average performance when evaluated with test time faults for 16 devices for SCMNIST

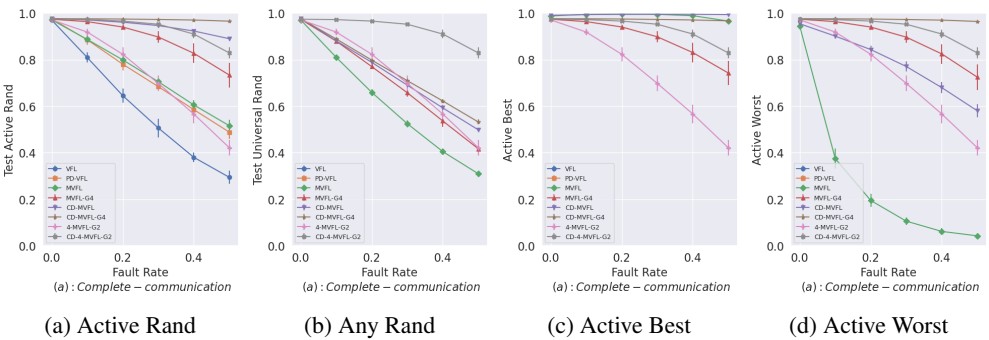

(a) Active Rand     (b) Any Rand     (c) Active Best     (d) Active Worst

Figure 14: All metrics reported for MNIST with 16 devices and only test time faults

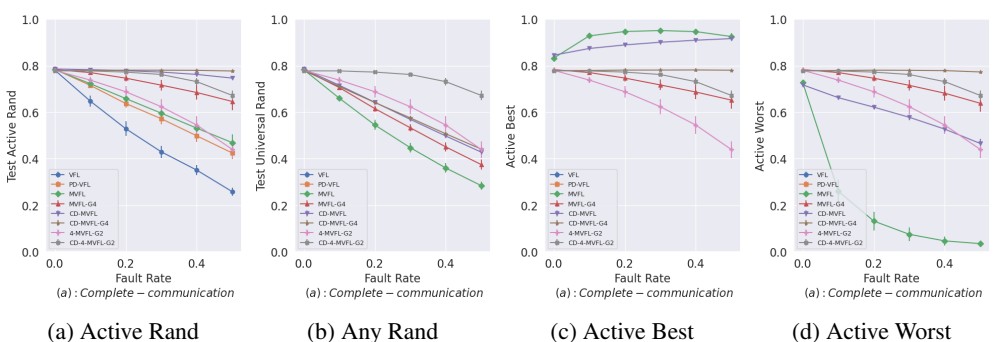

(a) Active Rand     (b) Any Rand     (c) Active Best     (d) Active Worst

Figure 15: All metrics reported for SCMNIST with 16 devices and only test time faults

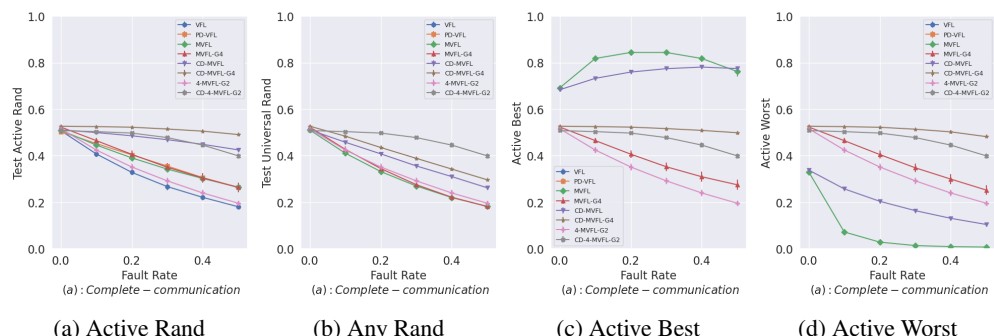

(a) Active Rand    (b) Any Rand    (c) Active Best    (d) Active Worst

Figure 16: All metrics reported for CIFAR-10 with 16 devices and only test time faults

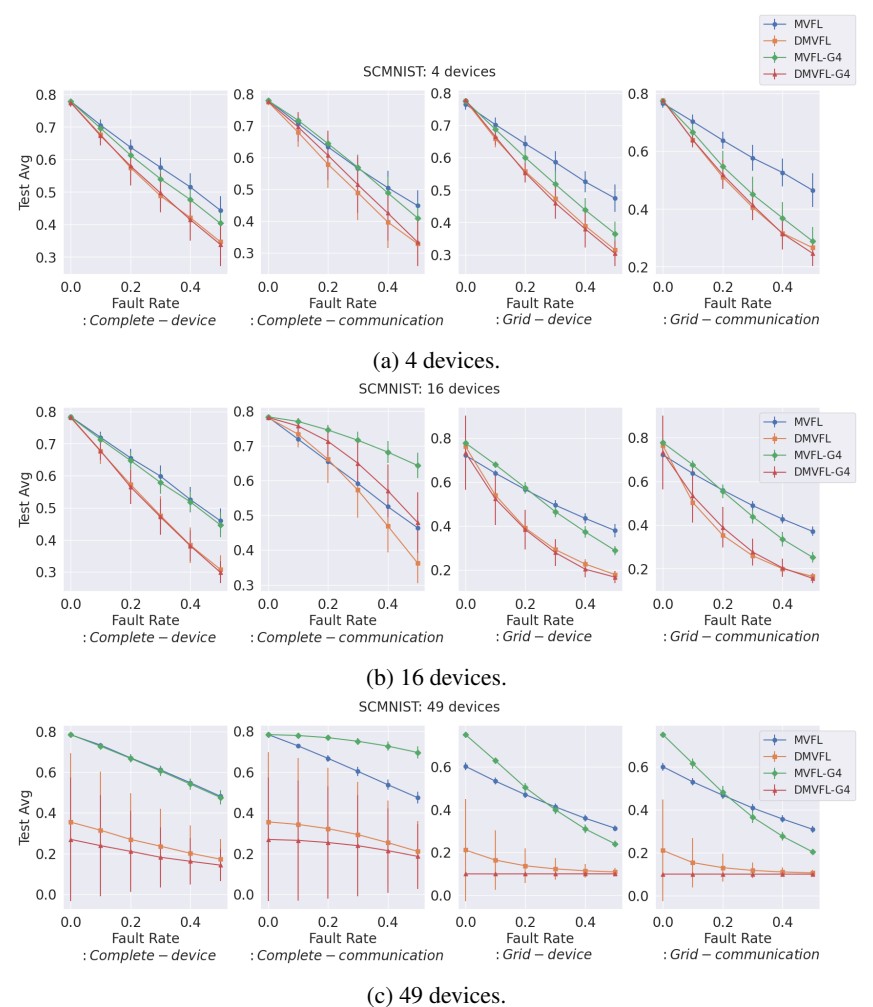

(a) 4 devices.

(b) 16 devices.

(c) 49 devices.

Figure 17: Test average accuracy for different test time fault rates for StarCraftMNIST with 4,16 and 49 devices. Observing the plots it can be concluded that VFL does not do well under different faulting conditions and MVFL or its gossip variant has the best performance.

a communication type of fault, gossiping does not improve the average performance. We believe this happens as a grid graph is quite sparse and training faults makes it more sparse. As a result, increasing gossip rounds does not lead to efficient passing of feature information from one client to another due to the sparseness, this is not the case in a complete graph. Furthermore, for reasons

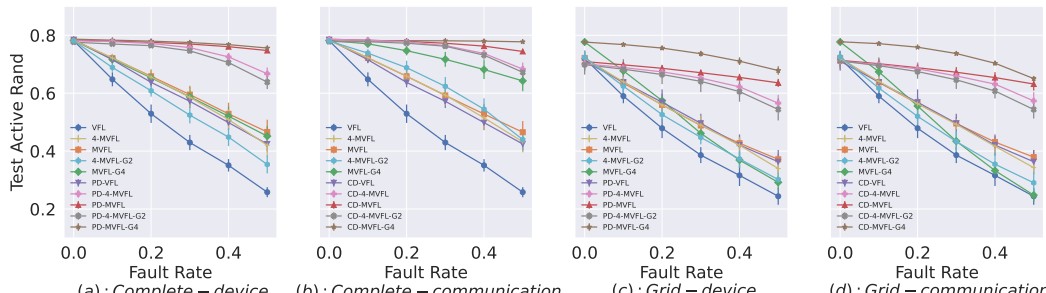

Figure 18: Test accuracy with and without communication (CD-) and party-wise (PD-) Drop out method for StarCraftMNIST with 16 devices. Here we include models trained under an dropout rate of 30% (marked by 'PD-' or 'CD-'). All results are averaged over 16 runs and error bar represents standard deviation. Across different configurations, MVFL-G4 trained with feature omissions has the highest average performance, while vanilla VFL performance is not robust as fault rate increases. As our experiments are repeated multiple times, what we report is the expectation (Avg) over the random active client selection.

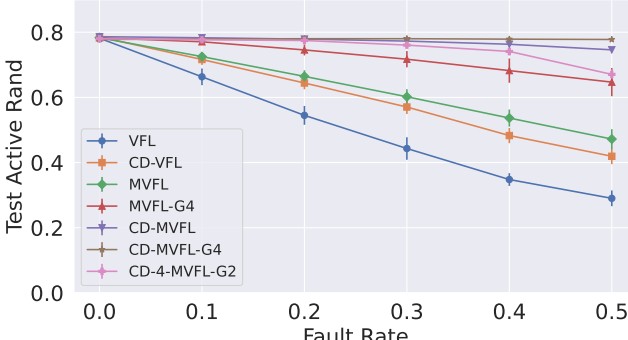

Figure 19: Test accuracy with and without communication (CD-) dropout method for StarCraftMNIST with 16 devices. Here we include models trained under an dropout rate of 30% (marked by 'CD-'). All results are averaged over 16 runs, and the error bar represents standard deviation. Across different configurations, MVFL-G4 trained with feature omissions has the highest average performance, while vanilla VFL performance is not robust as fault rate increases. As our experiments are repeated multiple times, what we report is the expectation (Avg) over the random active client selection.

mentioned in Section 4 of the main paper and Appendix G.5 of the Appendix, from Figure 12 we observe that adding any gossip rounds with device faults does not help in improving the performance. However, given the benefit 4 rounds of Gossip provides for Complete-Communication graph, we decided to use 4 Gossip rounds with MVFL.

We also investigated the effect of different number of Gossip rounds when using K-MVFL, in particular when the value of K is 4. Figure 13 (a) shows the the performance for different scenarios where dropout is not used during training and Figure 13 (b) shows for the condition such that dropout rate of 0.3% is used during training. It is observed that high number of Gossip rounds 4 is not having any significant benefit to performance and Gossip rounds of 0 and 2 are comparable in performance. Thus, including Gossip when the K=4 is not as beneficial as it was observed for the MVFL case. We conjecture that this likely happening because we are using gossip during training as well as during inference. We believe that using gossip only during inference and not during training will help

Table 4: Complete Communication Graph with 30% communication fault rate

| Number of aggregators | 1(VFL) | 2 | 4 | 16 |
|---|---|---|---|---|
| Active Rand (Avg) | 0.449 | 0.547 | 0.59 | 0.6 |
| # Comm. | 10.6 | 21 | 42 | 168.5 |

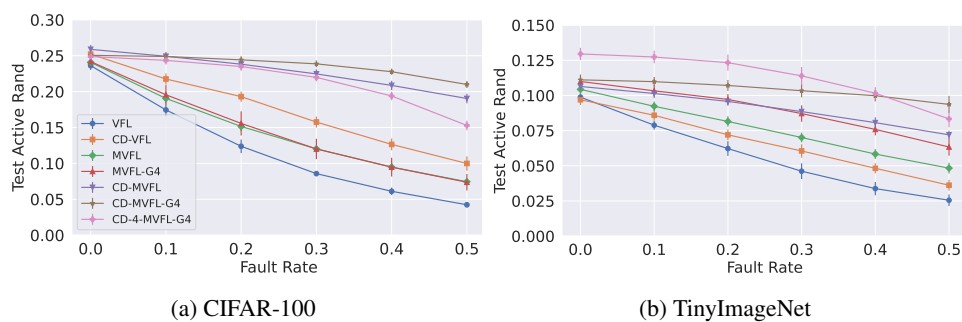

(a) CIFAR-100

(b) TinyImageNet

Figure 20: Accuracy for *complete-communication* test fault rates on CIFAR-100(a) and TinyImageNet(b) with 16 devices. The statistics are computed in the same manner as reported for Figure 2 in the main paper.

Table 5: Random Geometric Graph r=2.5 with 30% communication fault rate

| Number of aggregators | 1(VFL) | 2 | 4 | 16 |
|---|---|---|---|---|
| Active Rand (Avg) | 0.42 | 0.52 | 0.58 | 0.59 |
| # Comm. | 7.4 | 13.9 | 29 | 114.9 |

improve the performance with more gossip rounds. This, for this paper, when using 4-MVFL, we use 2 rounds of Gossip.

### G.5 EXPLORATION OF TEST FAULT RATES AND PATTERNS

In Figure 21 we present a more comprehensive representation of performance of different settings of MAGS. Figure 2 is a subset of Figure 21.

A note regarding gossiping is that mainly helps MVFL in the case of communication fault. We believe this is because in the device fault case, irrespective of number of gossip rounds, the representations from faulted device cannot be obtained. On the other hand, multiple gossip rounds in communication fault scenario has the effect of balancing out the lost representation at a client via neighboring connections. Switching to the grid baseline network, a major observation here is the degradation in the performance of both MVFL and MVFL-G4. We conjecture that in this case, clients can only directly communicate with neighboring clients, thus it's harder to get information from clients far away and extra communication leads to less benefit while the smaller network size and receiving more faulted representation become a bottleneck. Similarly, we notice that MVFL outperforms MVFL-G4 when fault rate is very high, as there is a much higher chance that the network is disconnected in comparison to complete baseline network. In short, we conclude that when trained with no faults, MVFL is overall the best model while gossiping helps except with high fault rates under the grid baseline network.

### G.6 EXTENSION OF COMMUNICATION AND PERFORMANCE ANALYSIS

In Table 7 we present the extension (performance metric is presented with Standard Deviation information) of Table 2, which is shown in the main paper. Here the results are presented such that gossip is used during inference only.

Table 6: Grid Graph with 30% communication fault rate

| Number of aggregators | 1(VFL) | 2 | 4 | 16 |
|---|---|---|---|---|
| Active Rand (Avg) | 0.386 | 0.449 | 0.492 | 0.491 |
| # Comm. | 2 | 3.99 | 7.98 | 33.5 |

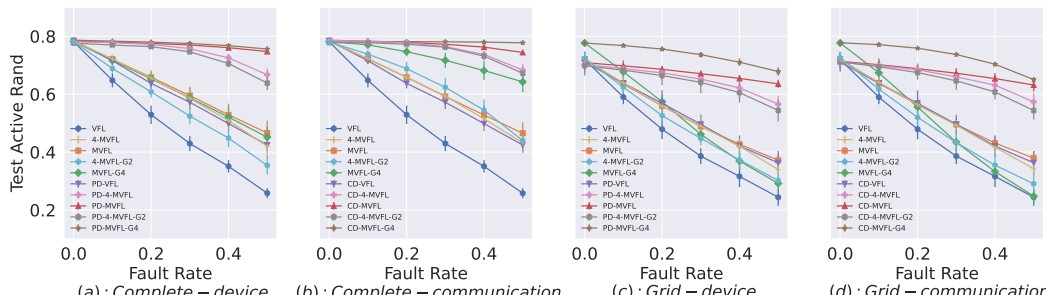

Figure 21: Test accuracy with and without communication (CD-) and party-wise (PD-) Dropout method for StarCraftMNIST with 16 devices. Here we include models trained under an dropout rate of 30% (marked by 'PD-' or 'CD-'). All results are averaged over 16 runs and error bar represents standard deviation. Across different configurations, MVFL-G4 trained with feature omissions has the highest average performance, while vanilla VFL performance is not robust as fault rate increases. As our experiments are repeated multiple times, what we report is the expectation (Avg) over the random active client selection.

Table 7: Active Rand (Avg) performance +/- 1 Std Dev at test time with 30 % communication fault rate. Compared to VFL, MVFL performs better but it comes at higher communication cost. Thus we propose 4-MVFL as a low communication cost alternative to MVFL. We want to highlight that 4-MVFL with poorly connected graph is still better than VFL with well connected graph, such as 4-MVFL with *RGG* (r=1) versus VFL with *Complete*.

| | Complete | | RGG r=2.5 | | RGG r=2 | | RGG r=1.5 | | RGG r=1 | | Ring | |
|---|---|---|---|---|---|---|---|---|---|---|---|---|
| | Avg | # Comm. | Avg | # Comm. | Avg | # Comm. | Avg | # Comm. | Avg | # Comm. | Avg | # Comm. |
| VFL | 0.430± 0.021 | 10.6 | 0.406± 0.032 | 7.4 | 0.407± 0.048 | 5.2 | 0.375± 0.043 | 3.5 | 0.386± 0.038 | 2.0 | 0.385± 0.036 | 1.4 |
| MVFL | 0.594±0.033 | 168.5 | 0.581±0.018 | 114.9 | 0.558±0.030 | 80.8 | 0.528±0.024 | 58.7 | 0.503±0.025 | 33.5 | 0.507±0.013 | 22.7 |
| 4-MVFL | 0.591±0.033 | 42 | 0.572±0.024 | 29 | 0.555±0.037 | 20.4 | 0.517±0.027 | 14.8 | 0.488±0.026 | 7.98 | 0.485±0.026 | 5.6 |
| MVFL-G4 | 0.732±0.017 | 836.2 | 0.728±0.032 | 572.1 | 0.721±0.026 | 407.2 | 0.689±0.038 | 293.9 | 0.62±0.029 | 168.2 | 0.558±0.023 | 113.4 |
| 4-MVFL-G2 | 0.687±0.027 | 126 | 0.661±0.044 | 87 | 0.623±0.052 | 61.2 | 0.566±0.041 | 44.8 | 0.491±0.034 | 23.94 | 0.484±0.046 | 16.8 |

## G.7 BEST, WORST AND SELECT ANY METRICS

In Figures 14 to 16 we present not only the Rand Active but also Rand Universal, Active Best and Active Worst metrics when evaluation are carried out for 16 Devices/Clients under only test faults. In the main paper, Table 3 is a subset of the comprehensive data presented here.

In addition, we also share Table 8 here, which aggregates information for an additional, 30% inference fault rate. While the table in the main paper shows data for only 50% fault rate.

## G.8 EVALUATION FOR DIFFERENT NUMBER OF DEVICES/CLIENTS

In Figure 17 we present average performance as a function of test time faults for three different number of devices. For all the different cases, it is observed that MVFL or its gossip variant performs the best. On observing the *Complete-Communication* plots for Figure 17, it can be seen that MVFL with gossiping has a more significant impact when the number of devices are 49 or 16 compared to when the number of devices are 4.

Table 8: Best models for 30% *complete-communication* test fault rate within 1 standard deviation are bolded. More detailed results with standard deviation are shown in the Appendix.

| | | MNIST | | | | SCMNIST | | | | CIFAR10 | | | |
| | | Active | | | Any | Active | | | Any | Active | | | Any |
| | | Worst | Rand | Best | Rand | Worst | Rand | Best | Rand | Worst | Rand | Best | Rand |
|---|---|---|---|---|---|---|---|---|---|---|---|---|---|
| | VFL | nan | 0.507 | nan | nan | nan | 0.430 | nan | nan | nan | 0.267 | nan | nan |
| | PD-VFL | nan | 0.684 | nan | nan | nan | 0.572 | nan | nan | nan | 0.355 | nan | nan |
| Fault Rate = 0.3 | 4-MVFL-G2 | 0.632 | 0.693 | 0.751 | 0.526 | 0.572 | 0.624 | 0.675 | 0.482 | 0.238 | 0.293 | 0.356 | 0.232 |
| | MVFL | 0.106 | 0.705 | **0.995** | 0.524 | 0.075 | 0.592 | **0.951** | 0.444 | 0.013 | 0.342 | **0.843** | 0.269 |
| | MVFL-G4 | 0.897 | 0.897 | 0.899 | 0.657 | 0.715 | 0.716 | 0.717 | 0.533 | 0.348 | 0.350 | 0.352 | 0.275 |
| | CD-4-MVFL-G2 | 0.944 | 0.951 | 0.969 | **0.714** | 0.745 | 0.765 | 0.784 | **0.589** | 0.438 | 0.472 | 0.528 | 0.372 |
| | CD-MVFL-G4 | **0.972** | **0.973** | 0.972 | 0.709 | **0.780** | **0.780** | 0.781 | 0.575 | **0.514** | **0.515** | 0.516 | **0.389** |

