# OpenReview forum: "Robust Decentralized VFL Over Dynamic Device Environment"
_ICLR.cc/2025/Conference — Submitted to ICLR 2025_

### Official Review · Reviewer_DssE · 2024-10-31

**Soundness:** 3
**Presentation:** 3
**Contribution:** 2
**Rating:** 5
**Confidence:** 3

**Summary:**

The algorithm presented in this paper is designed to address the robustness of Vertical Federated Learning (VFL) in dynamic network environments, especially when devices may fail due to faults caused by environmental factors or extreme weather. The algorithm is called MAGS (Multiple Aggregation with Gossip Rounds and Simulated Faults), which improves robustness by simulating faults during training, replicating the data aggregator, and utilizing gossip propagation rounds.

**Strengths:**

The MAGS algorithm proposed in this paper is mainly compared with several existing Vertical Federation Learning (VFL) methods, including traditional VFL, PD-VFL, MVFL and its variants such as MVFL-G4 and CD-MVFL-G4. These methods have different capabilities in dealing with device failures, especially in distributed environments such as edge computing where devices may fail due to a variety of reasons (such as extreme weather, power problems, etc.) to fail, which in turn affects the reliability and performance of the whole system. MAGS enables the system to maintain good performance in spite of high equipment failure rates by simulating failures and multiple rounds of rumor propagation.

**Weaknesses:**

The algorithm proposed in this paper has several disadvantages or limitations that need further clarification. The first one is due to the fact that MAGS needs to perform additional rounds of rumor propagation, which may incur more computational and communication overhead. In addition, there seems to be a limitation in the applicability of the MAGS algorithm, based on this paper it is known that although MAGS excels in dealing with high failure rates, it may not be advantageous compared to other VFL methods in the case of lower failure rates. How can a better algorithm be chosen for a specific equipment failure rate in a practical application? It seems that this paper does not provide quantitative selection criteria.

**Questions:**

1.In Algorithm 1 proposed in this paper, the message interaction between clients is based on the multi-round Gossip protocol. However it seems that Algorithm 1 will put higher requirements on communication and computation resource consumption compared to the single round Gossip usually used in decentralized optimization. Can the authors avoid using the multi-round Gossip protocol?
2.Another important topic of concern to the reviewer is that the paper does not quantitatively analyze the effect of the number of Gossip rounds on the performance of the algorithm under different network structures. It makes the results of this paper lack theoretical interpretability. In addition, can the authors further explore how to automatically adjust the optimal number of Gossip rounds according to different network structures?
3.Could the authors describe the differences and connections between the two types of failures leading to dynamic networks considered in the paper and the Byzantine attacks (malicious server) and denial-of-service attacks (communication link corruption) that are common in centralized or decentralized federated learning literature?
4.The MAGS approach is presented in the paper and its superiority is shown in the experiments. However, the theoretical analysis section could be more detailed as to why the MAGS approach improves robustness. It is recommended to add a mathematical derivation of the convergence analysis of the MAGS algorithm and how the algorithm enhances robustness by simulating faults, replicating data aggregators, and gossip propagation.
5.The effect of network structure on performance can be further explained, especially how MAGS performs under different network topologies, such as complete graph, ring, star, etc., and whether the parameters or design ideas need to be adjusted.
6.In accordance with this paper, MVFL is the optimal model when trained in a fault-free scenario. This means that in scenarios where high failure rates are not anticipated, MAGS may not be the most efficient approach. Can the authors discuss the reasons for that phenomenon? Can the algorithm be modified so that it achieves optimal performance under different failure rate conditions?

**Details Of Ethics Concerns:**

This paper designs robust algorithms for the Vertical Federated Learning problem in dynamic network environments, especially considering the situation when devices may fail due to environmental factors or extreme weather induced failures. Overall the topic of this paper is novel and interesting, but the reviewer still has a few questions and suggestions for further clarification by the authors. Please see the details in "Question".

---

> ### Author Response · Authors · 2024-11-22
>
> Reviewer DssE thank you for the review. Here we try to address the questions and follow up on the weaknesses
>
> **Weaknesses**
>
> > The algorithm proposed in this paper has several disadvantages or limitations that need further clarification. The first one is due to the fact that MAGS needs to perform additional rounds of rumor propagation, which may incur more computational and communication overhead.
>
> We realize that having all nodes act as aggregators could increase the communication cost. Thus, we develop K-MVFL as a low communication cost alternative to MVFL. Even when the total number of devices are 16, we observe per Table 2 that having K=4 offers a reasonable trade-off between communication and accuracy
>
> >In addition, there seems to be a limitation in the applicability of the MAGS algorithm, based on this paper it is known that although MAGS excels in dealing with high failure rates, it may not be advantageous compared to other VFL methods in the case of lower failure rates. How can a better algorithm be chosen for a specific equipment failure rate in a practical application? It seems that this paper does not provide quantitative selection criteria.
>
> From Figure 2 we see that variants of MAGS (PD-MVFL-G4 and CD-MVFL-G4) lead to equal if not superior performance when compared to VFL for low and high fault rates. Furthermore, MAGS does generalize to VFL setup at each device, in the event the base network is completely connected graph.
>
> **Questions**
>
> > In Algorithm 1 proposed in this paper, the message interaction between clients is based on the multi-round Gossip protocol. However it seems that Algorithm 1 will put higher requirements on communication and computation resource consumption compared to the single round Gossip usually used in decentralized optimization. Can the authors avoid using the multi-round Gossip protocol?
>
> In our proposed method, MAGS, to achieve the appropriate trade-off between performance and communication, we propose K-MFVL. In our experiments, as summarized in Table 2, even a value of K=4 for 16 devices, achieves robust performance while having only two gossip rounds. Per proposition 3 in the paper, having a few rounds of gossip helps with reducing the variability among device outputs, promoting better performance. Thus, not having any gossip rounds is detrimental to the performance.
>
> > Another important topic of concern to the reviewer is that the paper does not quantitatively analyze the effect of the number of Gossip rounds on the performance of the algorithm under different network structures. It makes the results of this paper lack theoretical interpretability. In addition, can the authors further explore how to automatically adjust the optimal number of Gossip rounds according to different network structures?
>
> The number of Gossip rounds is a hyperparameter. We do some experiments, as summarized in Figure 12 (Appendix), to select the number of gossip rounds. In a future research work, we can investigate further the interplay between gossip rounds and network structures.
>
> > Could the authors describe the differences and connections between the two types of failures leading to dynamic networks considered in the paper and the Byzantine attacks (malicious server) and denial-of-service attacks (communication link corruption) that are common in centralized or decentralized federated learning literature?
>
> The failure modes considered in our study are non malicious and random in nature, which is different from Byzantine attacks and denial-of-service attacks that have malicious intent at its core.

---

> ### Author Response · Authors · 2024-11-22
>
> > The MAGS approach is presented in the paper and its superiority is shown in the experiments. However, the theoretical analysis section could be more detailed as to why the MAGS approach improves robustness. It is recommended to add a mathematical derivation of the convergence analysis of the MAGS algorithm and how the algorithm enhances robustness by simulating faults, replicating data aggregators, and gossip propagation.
>
> While we extensively discussed and analyzed our choices in Section 3, we are happy to distill the information provided in Section 3 to outline the rationale for choosing the different strategies. A key challenge for DN-VFL is that test-time inference faults result in missing values and a corresponding distribution shift compared to the training distribution. A natural approach to address this is to simulate inference faults using dropout. Thus, this insight inspired us to use Party-wise Dropout (PD) and Communication-wise Dropout (CD) during the training process.
>
> Because we are in DN-VFL context, where any node can fail, a key problem in the conventional VFL setup is that there is a single point of failure, i.e., the single server or data aggregator. Thus, the server going down results in a catastrophic failure. Hence, we propose the use of all clients as data aggregators to introduce fault-tolerance via redundancy, which we call MVFL. MVFL can tolerate the failure of any node.
>
> While multiple data aggregators help avoid system-level failures, the performance of each data aggregator may be poor due to faults, which could result in overall high dynamic risk even if catastrophic failures are alleviated. Our key insight is to notice that the data aggregator models are diverse because each will have access to different client data and latent representations due to the graph topology and faults at test time. Because of this diversity, ensemble approaches improve the overall performance, as we prove in Sec. 3.   Thus, to implement ensembling in a distributed manner, we decided to use gossip layers to combine predictions among data aggregators. To study the effect of each of the components, the plots in Figure 2 are broken down into multiple components, as indicated by the legends. The progression from baseline VFL to our approach with each component added (i.e., CD-VFL, CD-MVFL, and CD-MVFL-G4) shows that each component improves the robustness of our final MAGS method.
>
>
> > The effect of network structure on performance can be further explained, especially how MAGS performs under different network topologies, such as complete graph, ring, star, etc., and whether the parameters or design ideas need to be adjusted.
>
> In Table 2 we study the performance of MAGS for different network topologies. To not mix things up, we did not change parameters like number of gossip rounds and number of aggregators across different graph types
>
> >In accordance with this paper, MVFL is the optimal model when trained in a fault-free scenario. This means that in scenarios where high failure rates are not anticipated, MAGS may not be the most efficient approach. Can the authors discuss the reasons for that phenomenon? Can the algorithm be modified so that it achieves optimal performance under different failure rate conditions?
>
> We have responded to this as a part of second point under weakness

---

> > ### Comment · Reviewer_DssE · 2024-12-01
> >
> > Thank you for responding to my review, and after reading your response and discussing it with other reviewers, I have decided to keep my score in view of the algorithm's overhead and application limitations.

---

### Official Review · Reviewer_8kJT · 2024-11-02

**Soundness:** 2
**Presentation:** 3
**Contribution:** 2
**Rating:** 3
**Confidence:** 3

**Summary:**

This paper attempts to improve the robustness of the (Vertical) Federated Learning framework, where data features are spread across different clients. It proposes a method called MAGS (Multiple Aggregation with Gossip Rounds and Simulated Faults) to overcome scenarios of device failures and communication reliability.  The authors propose combining three strategies to enhance fault tolerance and resilience: (1) fault simulations (dropout) during training, (2) aggregation redundancy using multiple clients as data aggregators, and (3) Gossip rounds for reducing prediction variability and improving resiliency.

The authors claim to formalize the integration of these strategies into a framework that they define as Dynamic Network Vertical Federated Learning (DN-VFL). They introduce a dynamic risk metric to assess model performance and propose three propositions. Proposition 1 establishes a lower bound on the dynamic risk of their approach. Proposition 2 highlights the benefits of averaging prediction using gossip rounds by comparing the risk of ensembled predictions across multiple clients versus that of a single prediction by an individual client. Proposition 3 claims to establish a bound on the difference between the ensemble and the local predictions.

Empirical experiments were conducted on various datasets, including MNIST and StarCraftMNIST, among several others. A key aspect of these studies was to evaluate the performance of the MAGS approach under device and communication faults. The authors compared performance with the Vanilla VFL approach and VFL variations, including party-wise and communication-wise dropouts. Consensus algorithms were also studied.

**Strengths:**

The paper attempts to formalize the combination of three strategies, which have traditionally been used in isolation, to improve the robustness of the VFL framework. The authors formalize some aspects of their approach and performs a reasonably comprehensive experimental study.

**Weaknesses:**

My primary concern with this work is that, by allowing peer-to-peer communication between devices, the framework diverges from the original design and motivation of Vertical Federated Learning. VFL was specifically devised for environments where direct communication between parties is impractical or restricted due to privacy, regulatory, or technical limitations. By introducing peer-to-peer communication through gossip rounds, client-based aggregation, and ensembling, the proposed framework theoretically violates the foundational principles of federated learning. In fact, the framework now resembles a multi-agent system, which allows direct interactions among agents to achieve consensus.  This seems to be a topic that is closely related and yet there is no mention of it in the literature review.  Can the authors explain how their MAGS framework maintains the privacy guarantees of VFL while introducing peer-to-peer communication? Are there specific use cases where this trade-off is justified?

Proposition 1 highlights risk reduction when using multiple aggregators, which seems to be an intuitive results considering that the MAGS framework increases redundancy. Similarly, Proposition 2 highlights a reduction in variance which is a natural byproduct of ensembling. Proposition 3 also seems to be a simple and direct adaptation of a well-known consensus result in distributed averaging. Could the authors clarify how these propositions provide new insights or guarantees specific to the DN-VFL setting that are not immediately obvious from existing results in other domains?

**Questions:**

Can the authors explain how their MAGS framework maintains the privacy guarantees of VFL while introducing peer-to-peer communication? Are there specific use cases where this trade-off is justified?

Could the authors clarify how these propositions provide new insights or guarantees specific to the DN-VFL setting that are not immediately obvious from existing results in other domains?

Could the authors explain how MAGS differs from a multi-agent system (MAS)? While MAS commonly involves multiple agents collaborating for decision-making, similar configurations can also be applied to Machine Learning. Can authors discuss any specific advantages that MAGS might have over traditional MAS approaches in this context? Alternatively, are there any VFL-specific constraints that MAGS addresses that a general MAS approach might not?

---

> ### Author Response · Authors · 2024-11-22
>
> Reviewer 8kJT thank you for the review. Here we try to address the questions and follow up on the weaknesses
>
> **Weaknesses**
>
> > My primary concern with this work is that, by allowing peer-to-peer communication between devices, the framework diverges from the original design and motivation of Vertical Federated Learning. VFL was specifically devised for environments where direct communication between parties is impractical or restricted due to privacy, regulatory, or technical limitations. By introducing peer-to-peer communication through gossip rounds, client-based aggregation, and ensembling, the proposed framework theoretically violates the foundational principles of federated learning. In fact, the framework now resembles a multi-agent system, which allows direct interactions among agents to achieve consensus. This seems to be a topic that is closely related and yet there is no mention of it in the literature review. Can the authors explain how their MAGS framework maintains the privacy guarantees of VFL while introducing peer-to-peer communication? Are there specific use cases where this trade-off is justified?
>
> In the first paper on Federated Learning by Google researchers [1], federated learning is defined as a general term used for distributed machine learning over heterogeneous data. While privacy is a distinct advantage of using Federated Learning, privacy in itself is not stated as a necessary condition for using federated learning.
>
> Furthermore, while privacy is critical for some applications (e.g., healthcare) it is not the primary objective for machine generated IoT/Sensor Network applications that our paper is primarily concerned with. Our method is geared towards attaining robust performance in situations of extreme faults. As mentioned in the paper on line 34 the type of use case we are targeting are safety-critical applications such as search and rescue in underground mines, where an intelligent device network needs to continue operating even under near catastrophic faults (e.g., 50% of devices fail). In addition, orthogonal privacy preserving approaches such as blockchain-based [2] and homomorphic encryption techniques [3] can be used in conjunction with our method. Finally, in Appendix E (line 1026) of the paper, we highlight that our focus is not on privacy since we assume a trusted, though unreliable network of devices and our contributions are orthogonal and complementary to advancements in VFL privacy.
>
> [1] McMahan, B., Moore, E., Ramage, D., Hampson, S. and y Arcas, B.A., 2017, April. Communication-efficient learning of deep networks from decentralized data. In Artificial intelligence and statistics (pp. 1273-1282). PMLR.
> [2] Li, S., Yao, D. and Liu, J., 2023, July. Fedvs: Straggler-resilient and privacy-preserving vertical federated learning for split models. In International Conference on Machine Learning (pp. 20296-20311). PMLR.
> [3] Tran, L., Chari, S., Khan, M.S.I., Zachariah, A., Patterson, S. and Seneviratne, O., 2024, July. A differentially private blockchain-based approach for vertical federated learning. In 2024 IEEE International Conference on Decentralized Applications and Infrastructures (DAPPS) (pp. 86-92). IEEE.
>
> > Proposition 1 highlights risk reduction when using multiple aggregators, which seems to be an intuitive results considering that the MAGS framework increases redundancy. Similarly, Proposition 2 highlights a reduction in variance which is a natural byproduct of ensembling. Proposition 3 also seems to be a simple and direct adaptation of a well-known consensus result in distributed averaging. Could the authors clarify how these propositions provide new insights or guarantees specific to the DN-VFL setting that are not immediately obvious from existing results in other domains?.
>
> Our approach seeks to integrate insights from various subfields (resilient systems, ensembling, and consensus algorithms) to establish a theoretical foundation for the design decisions made in MAGS. We do not claim to introduce entirely new theoretic concepts; instead, our goal is to apply existing theory to our specific context to enhance its theoretical grounding. Although this might appear intuitive and self-evident after the fact, these results give theoretic reasons for why our proposed approach actually works, and we argue that this would not be obvious to all readers.

---

> > ### Author Response · Authors · 2024-11-22
> >
> > **Questions**
> >
> > >Can the authors explain how their MAGS framework maintains the privacy guarantees of VFL while introducing peer-to-peer communication? Are there specific use cases where this trade-off is justified?
> >
> > We have addressed this above under Point 1 in relation to the identified weakness.
> >
> > >Could the authors clarify how these propositions provide new insights or guarantees specific to the DN-VFL setting that are not immediately obvious from existing results in other domains?
> >
> > We have addressed this above under Point 2 in relation to the identified weakness.
> >
> > >Could the authors explain how MAGS differs from a multi-agent system (MAS)? While MAS commonly involves multiple agents collaborating for decision-making, similar configurations can also be applied to Machine Learning. Can authors discuss any specific advantages that MAGS might have over traditional MAS approaches in this context? Alternatively, are there any VFL-specific constraints that MAGS addresses that a general MAS approach might not?
> >
> > We are not quite sure what you mean by MAS and “traditional MAS approaches”. Could you explain more what you mean? While we see some high-level resemblance, MAS is a very large field, and it is unclear what you mean.
> >
> > From our understanding, MAS is more closely related to reinforcement learning, where only zero-order information can be exchanged, each agent can have different objectives and the environment can change over time depending on the actions taken. Thus, the context difference is similar to supervised learning vs reinforcement learning. In our context, we are focused on a supervised learning setup. In the MAGS context, we assume that all “agents” are collaborative and aiming towards the same goal. We do not allow the agents to make any “decisions” as the algorithm is fixed.

---

> ### Comment · Reviewer_8kJT · 2024-11-25
>
> Thank you for responding to my comments. MAS, or Multi-Agent Systems, is a prominent field in collaborative decision-making that emphasizes decentralized coordination. It primarily relies on peer-to-peer communication between agents to share information and achieve collective goals. I am keeping my score.

---

### Official Review · Reviewer_a5wU · 2024-11-03

**Soundness:** 3
**Presentation:** 2
**Contribution:** 3
**Rating:** 3
**Confidence:** 4

**Summary:**

This paper studies decentralized vertical federated learning over a dynamic device environment, which the authors define as faults in changing client graphs. The authors develop a novel DN-VFL that brings together dropout, replication, and gossiping and show its outperformance over baselines via empirical results.

**Strengths:**

* The paper identifies an important issue in the VFL setup, where fault can occur when the clients or the communications betwwen different clients stochastically drop out.
* The paper provides extensive experiments on different datasets.
* The design of algorithms is intuitive, and the derivations seem correct.

**Weaknesses:**

* The most obvious weakness is the presentation of the paper. The authors should try to shorten the overview of each section and go straight to the point. For example, the overview of Section 3 is just a wordy replication of what is coming next. It might be helpful to use bullet points to highlight the contributions.
* The related work is not thorough enough. Except asynchronous decentralized learning, the authors have to consider a topic called client unavailability problem in federated learning, where clients may (stochastically) drop out, e.g., in [1,2,3]. Specifically, they should talk about the connections and differences between them and this work.
* There are many contents that seem to be out of the space in Section 2. For instance, different client selection strategies have been talked about in great detail, which should belong to the development of algorithms instead.
* Following the problem formulation, there is no motivating example to illustrate that a dynamic network can lead to significant performance degradation to the classic VFL algorithms. Although it might be intuitive to conjecture that is the case, evidence is needed.
* Some of the definitions are missing. In Section 3.1, the authors talk about adopting Communication-wise Dropout. Yet, no formal definitions are given.
* It is hard to parse the different components directly from Algorithm 1. No elaborations on Algorithm 1 have been given; neither do the authors talk about the connections in Sections 3.2 and 3.3.
* The most confounding part is that the authors greatly emphasize the case when there are no active aggregators. In that case, a trivial solution is to skip and move on to the next round. In addition, Proposition 1 is a lower bound, meaning that the second term, which depends on the probability of an empty active client set, can be ignored.
* In Proposition 2, it is also unclear what the dynamic risk of an ensemble leads to in the sense that no matching upper bound for risk with faults is proved.







**References.**
[1] Gu, X., Huang, K., Zhang, J., & Huang, L. (2021). Fast federated learning in the presence of arbitrary device unavailability. Advances in Neural Information Processing Systems, 34, 12052-12064.

[2] Wang, S., & Ji, M. (2023). A Lightweight Method for Tackling Unknown Participation Statistics in Federated Averaging. arXiv preprint arXiv:2306.03401.

[3] Wang, S., & Ji, M. (2022). A unified analysis of federated learning with arbitrary client participation. Advances in Neural Information Processing Systems, 35, 19124-19137.

**Questions:**

* Can the authors provide numerical or theoretical justifications that the dynamic device environment will lead to detrimental effects?
* What is the difference between Definition 2 and Definition 3? It seems that Definition 3 subsumes Definition 2 although with different probabilities.
* Which definition (2 or 3) do the authors consider when proving their theoretical results?
* Can the authors provide formal definitions of communication-wise dropout?
* Can the authors outline the connections between the pseudocode Algorithm 1 and their development of methods?
* Regarding experiment setups, the authors compare their serverless methods with VFL methods with a server while also including baseline communication networks. Can authors elaborate on the experiment setup for VFL methods with a server?
* Why are there NAN results in Table 3 and Table 8 (in Appendix) for VFL and PD-VFL? Can the authors elaborate the difficulty in not being able to produce meaningful results for VFL and PD-VFL?

---

> ### Author Response · Authors · 2024-11-22
>
> Reviewer a5wU thank you for the review. Here we try to address the questions and follow up on the weaknesses
>
> **Weaknesses**
> >The most obvious weakness is the presentation of the paper. The authors should try to shorten the overview of each section and go straight to the point. For example, the overview of Section 3 is just a wordy replication of what is coming next. It might be helpful to use bullet points to highlight the contributions.
>
> Thank you for the feedback. At the beginning of each section our goal was to give a brief summary and then dive into the details. We can certainly consider trimming down the sections. [In progress and will incorporate the edits in the paper and upload it]
>
> >The related work is not thorough enough. Except asynchronous decentralized learning, the authors have to consider a topic called client unavailability problem in federated learning, where clients may (stochastically) drop out, e.g., in [1,2,3]. Specifically, they should talk about the connections and differences between them and this work
>
> Thank you for pointing these papers out. These papers are related to horizontal federated learning (HFL) and thus we did not put much emphasis on these given our context is that of vertical federated learning (VFL). As mentioned in one of the responses above (Qns 1 by Reviewer QXxf), why VFL and HFL are different and thus the HFL related methods mentioned in [1,2,3] are not directly applicable to our context.
>
> >There are many contents that seem to be out of the space in Section 2. For instance, different client selection strategies have been talked about in great detail, which should belong to the development of algorithms instead
>
> We discuss client selection in Section 2 because we aim to align the metrics with real-world scenarios rather than treating them as decisions made by the algorithm. Each metric represents a distinct real-world context. For instance, the "Select Any Client" metric simulates a scenario where an external entity queries a specific device without knowing whether it can transmit its output. Thus, the different metrics using different client selection strategies properly belong in the problem setup and not in the methodology section.
>
> > Following the problem formulation, there is no motivating example to illustrate that a dynamic network can lead to significant performance degradation to the classic VFL algorithms. Although it might be intuitive to conjecture that is the case, evidence is needed.
>
> In Figure 2 we provide data to show that performance of VFL decreases significantly at higher fault rates for both device and communication faults. For instance, in a complete communication graph the performance of VFL drops from roughly 80% at 0% fault rate to approximately 25% at 50% fault rate for both communication and device faults. These results do make sense as the active party/server in VFL requires representations from other passive parties for inference. However, if the passive parties are unable to share information due to communication or device faults, the overall performance of the VFL system will suffer. For instance, if we consider a sensor network monitoring a region, then if due to faults a large portion of the sensors are not able to communicate their information, the global performance of such a system will not be good due to missing information.
>
>
> > Some of the definitions are missing. In Section 3.1, the authors talk about adopting Communication-wise Dropout. Yet, no formal definitions are given.
>
> On line 296 we define  Communication-wise Dropout (CD), but do not dedicate a definition block to it.
>
> >It is hard to parse the different components directly from Algorithm 1. No elaborations on Algorithm 1 have been given; neither do the authors talk about the connections in Sections 3.2 and 3.3.
>
> Thank you for the feedback. We can annotate in Algorithm 1 the relevant sections 3.2 and 3.3 accordingly.[In progress and will incorporate the edits in the paper and upload it].

---

> > ### Author Response · Authors · 2024-11-22
> >
> > > The most confounding part is that the authors greatly emphasize the case when there are no active aggregators. In that case, a trivial solution is to skip and move on to the next round. In addition, Proposition 1 is a lower bound, meaning that the second term, which depends on the probability of an empty active client set, can be ignored.
> >
> > While we understand your concern, we believe the no active aggregators part is still important for a robust method in practice (though perhaps only a few aggregators are needed to avoid this). From what we understand, we think you mean that if devices fail and recover every round, you could just wait until at least one aggregator recovers and then perform inference thereby ignoring the last term. We argue that this is not always a feasible option. In reality, devices that fail may take a long time to recover since they may require manual intervention or certain conditions like sunny weather. Thus, if there is a single aggregator and it fails, the inference latency could be arbitrarily long given your solution. We expect that real devices could have longer failures that would make a single aggregator a major problem.
> >
> >
> > > In Proposition 2, it is also unclear what the dynamic risk of an ensemble leads to in the sense that no matching upper bound for risk with faults is proved.
> >
> > We do not fully understand your concern. Our main claim is that the ensemble risk will always be less than the non-ensemble risk if there is any diversity in the predictions, regardless of faults. We do not claim that the ensemble approach is optimal w.r.t. all possible approaches, i.e., it doesn’t necessarily achieve the optimal risk with faults. While it would be nice to have, we do not believe a matching upper bound on the risk should be required for our paper.
> >
> > **Questions**
> >
> > > Can the authors provide numerical or theoretical justifications that the dynamic device environment will lead to detrimental effects?
> >
> > Responded as a part of 4th weakness mentioned above
> >
> > >What is the difference between Definition 2 and Definition 3? It seems that Definition 3 subsumes Definition 2 although with different probabilities
> >
> > Definition 2 defines what a device fault means in a network and Definition 3 defines what a communication fault refers to in a network. Thus, these two define two different conditions
> >
> >
> > >Which definition (2 or 3) do the authors consider when proving their theoretical results?
> >
> > Both are referring to different things. For device faults (Definition 2 is used), for communication faults (Definition 3 is used). For proposition 1, Definition 2 is used.
> >
> > >Can the authors provide formal definitions of communication-wise dropout?
> >
> > On line 296, we define Communication-wise Dropout (CD) as applying dropout to client-to-client communication instead of just client-to-server communication.
> >
> > >Can the authors outline the connections between the pseudocode Algorithm 1 and their development of methods?
> >
> > Responded as a part of 6th weakness mentioned above
> >
> > >Regarding experiment setups, the authors compare their serverless methods with VFL methods with a server while also including baseline communication networks. Can authors elaborate on the experiment setup for VFL methods with a server?
> >
> > Our VFL method with a server is a standard VFL setup as defined in [1]. In this setup, an active party acts as a data aggregator, often referred to as a server and all the other devices/entities act as passive parties, which send data to the active party/server. Since this is a well established VFL setup, we did not elaborate on it in our paper. We should note that in our DN-VFL context the "server"/active party is not special and can also fault. Thus the server in our setup does not have any special ability.
> >
> > >Why are there NAN results in Table 3 and Table 8 (in Appendix) for VFL and PD-VFL? Can the authors elaborate the difficulty in not being able to produce meaningful results for VFL and PD-VFL?
> >
> > In the VFL setup there is only one data aggregator and concepts like “Worst” and “Best” are not applicable, as they are used when there are multiple data aggregators.
> >
> > [1] Yang Liu, Yan Kang, Tianyuan Zou, Yanhong Pu, Yuanqin He, Xiaozhou Ye, Ye Ouyang, Ya-Qin Zhang, and Qiang Yang. Vertical federated learning. arXiv preprint arXiv:2211.12814, 2022.

---

> > > ### Comment · Reviewer_a5wU · 2024-11-26
> > >
> > > Thank the authors for their responses. After reading them and the other reviewers' comments, I decided to keep my score.
> > >
> > > For example, it remains unclear whether the algorithm implementations from Algorithm 1 can be parsed directly, nor did the authors provide pointers to external equations or relevant sections. This is also echoed by reviewer QXxf's comment on the definitions of "aggregator-specific risk", which I also failed to find or understand.

---

### Official Review · Reviewer_QXxf · 2024-11-07

**Soundness:** 3
**Presentation:** 2
**Contribution:** 2
**Rating:** 6
**Confidence:** 3

**Summary:**

This work considers the problem of collaborative inference in vertical federated learning (VFL). In VFL, all clients have the same datapoints, but different subsets of features for each datapoint. Faults in VFL can disrupt both training and inference due to the need for client communication during inference. This work formalizes the problem statement of faults in VFL -- both during the computation in a node, and during communications between nodes.

**Strengths:**

The paper proposes the notion of dynamic risk, where the expectation of loss is also computed over the stochasticity of faults. In the proposed algorithm, the language model heads of the models at each node are trained by simulating faults via dropouts -- which introduces robustness to faults during test time. The work also proposes several heuristics such as multiple VFL, where the data aggregator is replicated to prevent catastrophic faults with a single point-of-failure. Moreover, gossip rounds o ensemble the predictions from multiple data aggregators reducing the prediction variance across devices. (Unlike vanilla VFL, in DN-VFL, the clients are allowed to act as data aggregators and communicate with each other). Inference is done in two steps: Firstly, each client makes a prediction. This is followed by an "entity" collection these predictions, and subsequently making a final prediction.

This approach seems like a sound and well-defined approach to the problem of faults in VFL.

**Weaknesses:**

My major concern (which prevents me from giving a higher score) is that it is not completely clear to me what are the technical challenges that were overcome and novelty of the proposed approach. It majorly seems like some orthogonal approaches are combined to give the proposed approach. While this is not a negative thing per se, it would be highly appreciated if some discussions were added regarding the choices made, which will help clarify some things. For example:

1. Multiple data aggregators have been used in prior works with conventional federated learning too. It falls under the umbrella of "semi-decentralized federated learning" Prior works consider communication failures between data aggregators as well. Is there any reason why it was not considered in this work? This can perhaps be mitigated by using multiple rounds of gossip, and/or optimizing the collaboration weights during the data aggregation process.

2. It was not clear to me -- is the idea of training classifier heads with dropout to simulate test-time faults novel to this paper, and has not been considered in prior work? If so, how scalable is this approach, i.e., is it possible to train the classifier heads for robustness by simulating faults on very large networks (or must it be done in a clustered fashion)?

**Questions:**

In addition, I have some more clarification questions:

1. Proposition statements need to be more precise -- eg., in Prop. 1, what is "aggregator specific risk"?

2. More implementation details are needed -- what is the predictor used at each client?

I would be more than happy to reconsider my score contingent on other reviews, and the authors' responses (especially the concerns in the Weaknesses section).

**Details Of Ethics Concerns:**

None needed.

---

> ### Author Response · Authors · 2024-11-22
>
> Reviewer QXxf thank you for the review. Here we try to address the questions and follow up on the Weaknesses
>
> **Weaknesses**
> > My major concern (which prevents me from giving a higher score) is that it is not completely clear to me what are the technical challenges that were overcome and novelty of the proposed approach.
>
> Our first novelty lies in formalizing a new ML robustness problem for decentralized vertical federated learning (VFL) (note this is fundamentally different from horizontal FL). Our framework formalizes the problem as loss under a dynamic network model and introduces the dynamic network risk, carefully handling communication to an external entity. This new robustness problem is analogous to formalizing out-of-distribution robustness, such as domain generalization, domain adaptation, and adversarial robustness. Our new robustness problem gives rise to three specific technical challenges (as summarized in Table 1):
> 1. *Server/active party faults* - Prior VFL works did not consider this case likely because they focus on the cross-silo setting. Yet in the cross-device setting, this seems like a very natural scenario.
> 2. *Decentralized setup with faults* - While some prior works have considered decentralized FL, no VFL works consider both a decentralized and faulting setup at the same time.
> 3. *Faults at inference* - While prior VFL works considered faults during training, few works focused on faults at VFL inference (note: HFL need not consider faults at inference).
>
> Inspired by our new robustness formalization and its corresponding technical challenges, our second novelty is bringing together techniques from different areas in order to solve this problem. While we acknowledge that each component (drop-out, multiple data aggregators, gossip) when taken alone is not a novel contribution (and that our main novelty is in our robustness problem formalization), it is not a priori obvious that the combination of these three components will build upon each other for a practically robust method. Yet, indeed, we analyze each component for its contribution to robustness and show empirically that each of these are necessary for robustness. Finally, we would like to emphasize that the best solutions (particularly for new problems) are often relatively simple in hindsight; yet, this should not be counted against an approach.
>
>
> > It majorly seems like some orthogonal approaches are combined to give the proposed approach. While this is not a negative thing per se, it would be highly appreciated if some discussions were added regarding the choices made, which will help clarify some things. For example:
>
> > 1. Multiple data aggregators have been used in prior works with conventional federated learning too. It falls under the umbrella of "semi-decentralized federated learning" Prior works consider communication failures between data aggregators as well. Is there any reason why it was not considered in this work? This can perhaps be mitigated by using multiple rounds of gossip, and/or optimizing the collaboration weights during the data aggregation process.
>
> The majority of work in the past related to  "semi-decentralized federated learning" has been done in the horizontal federated learning (HFL) domain. Limited attention has been given to semi-decentralized vertical federated learning (VFL) that considers failure of communication or dropping of any node. Unlike HFL, which involves averaging model parameters across clients and each client having access to the averaged model, VFL involves aggregating model outputs from different clients. As a result, during test time, a client in HFL makes inference which is independent of other clients. On the other hand, in a VFL setup, all clients participate to make an inference. Also, during test time, a server fault in HFL does not affect the capability of a client
> to make inference as the clients have a model. In addition, under this setting, a client fault has no effect on inference capability of other functional clients. On the contrary, for the VFL setup a server failure during inference results in the system being unable to make an inference.
>
> Furthermore, client faults in VFL affect the test time inference because under faulting scenario features may be lost during inference, even on a single sample. For instance if during test time a client faults, the features associated with that device (across all the samples), will not be able to participate in the process. Thus, leading to a loss in performance, especially if the features that are unable to participate due to faults are critical. Due to these unique challenges, HFL methods for handling faults cannot be applied to VFL inference and even adapting fault-tolerant HFL methods is non-trivial, given the difference between HFL and VFL.

---

> ### Author Response · Authors · 2024-11-22
>
> >2. It was not clear to me - is the idea of training classifier heads with dropout to simulate test-time faults novel to this paper, and has not been considered in prior work? If so, how scalable is this approach, i.e., is it possible to train the classifier heads for robustness by simulating faults on very large networks (or must it be done in a clustered fashion)?
>
> We want to first clarify that we do not incorporate dropout in the classifier heads, rather dropouts are simulating the loss in communication among devices. The idea of simulating test-time faults has been considered in prior works such as Party-wise Dropout (PD). However, since DN-VFL operates in a decentralized environment where clients communicate with each other, PD is insufficient. Hence, we propose Communication-wise Dropout.
>
> Conceptually, we do not anticipate challenges with scaling MAGS, our method. In Appendix G.8 (Figure 17) of the paper we have presented results with different numbers of devices 4,16,49 for Completely connected graphs under various inference fault rates. The trends observed suggest that even with scaling of devices, MVFL and its Gossip variant are having a better performance than VFL in the DN-VFL context.
>
>
> **Questions**
>
> >Proposition statements need to be more precise - eg., in Prop. 1, what is "aggregator specific risk"
>
> Sure, we are adding some more details to our paper and will upload it over the next few days. In Prop. 1 “aggregator specific risk” is referring to the Dynamic Risk at each client.
>
> >More implementation details are needed - what is the predictor used at each client?
>
> The predictor at each client is modeled by linear layers, as mentioned in Appendix F.3.

---

> > ### Comment · Reviewer_QXxf · 2024-11-30
> > **Ack for rebuttal + Increasing score**
> >
> > Thank you for your response.
> >
> > I have read the authors' rebuttal, and agree with them on most parts. I appreciate the authors' remark acknowledging prior works and not claiming superfluous novelty. Indeed, simple approaches applied to a new problem setting is interesting. I have decided to increase my score, and if needed, would be happy to discuss more post discussion period.

---

### Author Response · Authors · 2024-11-22
**Response to all**

We sincerely thank the reviewers for their valuable feedback. Reviewer QXxf found our approach to be sound and well-defined for addressing the problem of faults in VFL. Reviewers a5wU and 8kJT appreciated the extensiveness of our experimentation. Additionally, Reviewer a5wU noted that we have highlighted a critical issue in the VFL setting and acknowledged the correctness of our derivations. In the sections below we try to address the points that were raised by each reviewer.

---

### Author Response · Authors · 2024-11-26
**Uploaded PDF with modifications**

We uploaded an updated version of the paper with the following modifications:

- Updated Proposition 1. Instead of using “aggregator specific risk” we denote it by “risk of a predictor” and this is also highlighted in equation 1

- We trimmed down primarily the overview to section 3 and highlighted the contribution of this section using bold letters.

- In section 3, we referenced Algorithm 1 in section 3.2 and 3.3

All the new additions in the paper are highlighted in blue.

---

### Comment · Area_Chair_yr9J · 2024-11-26
**Response**

Dear Reviewers,

The authors have provided their rebuttal to your questions/comments. It will be very helpful if you can take a look at their responses and provide any further comments/updated review, if you have not already done so.

Thanks!

---

### Meta-Review · Area_Chair_yr9J · 2024-12-20

**Metareview:**

This paper considers a vertical federated learning setting, distributed training with features distributed across compute nodes, where nodes may fail. The reviewers complained about clarity and discussions of relevant literature. There are also questions regarding effectiveness of the proposed system, which persists after the discussions.

Based on the reviews and comments, I recommend rejection.

**Additional Comments On Reviewer Discussion:**

The reviewers have participated in discussion and are overall unconvinced about the papers.

---

### Decision · Program_Chairs · 2025-01-22

Reject